# Role of FAM134 paralogues in endoplasmic reticulum remodeling, ER-phagy, and Collagen quality control

Alessio Reggio[1] (ID), Viviana Buonomo[1,†], Rayene Berkane[2,3,†], Ramachandra M Bhaskara[2,3,4] (ID), Mariana Tellechea[2,3,5], Ivana Peluso[1], Elena Polishchuk[1], Giorgia Di Lorenzo[1], Carmine Cirillo[1], Marianna Esposito[1], Adeela Hussain[6], Antje K Huebner[6], Christian A Hübner[6], Carmine Settembre[1] (ID), Gerhard Hummer[4,7] (ID), Paolo Grumati[1,*] (ID) & Alexandra Stolz[2,3,**] (ID)

## Abstract

Degradation of the endoplasmic reticulum (ER) via selective autophagy (ER-phagy) is vital for cellular homeostasis. We identify FAM134A/RETREG2 and FAM134C/RETREG3 as ER-phagy receptors, which predominantly exist in an inactive state under basal conditions. Upon autophagy induction and ER stress signal, they can induce significant ER fragmentation and subsequent lysosomal degradation. FAM134A, FAM134B/RETREG1, and FAM134C are essential for maintaining ER morphology in a LC3-interacting region (LIR)-dependent manner. Overexpression of any FAM134 paralogue has the capacity to significantly augment the general ER-phagy flux upon starvation or ER-stress. Global proteomic analysis of FAM134 overexpressing and knockout cell lines reveals several protein clusters that are distinctly regulated by each of the FAM134 paralogues as well as a cluster of commonly regulated ER-resident proteins. Utilizing pro-Collagen I, as a shared ER-phagy substrate, we observe that FAM134A acts in a LIR-independent manner and compensates for the loss of FAM134B and FAM134C, respectively. FAM134C instead is unable to compensate for the loss of its paralogues. Taken together, our data show that FAM134 paralogues contribute to common and unique ER-phagy pathways.

**Keywords** autophagy; Collagen; ER stress; ER-phagy; FAM134
**Subject Categories** Autophagy & Cell Death; Organelles

## Introduction

The endoplasmic reticulum (ER) comprises the largest endomembrane system within eukaryotic cells and is responsible for a plethora of cellular functions including lipid and protein synthesis, protein quality control, ion homeostasis and inter-organelle communication (Chen *et al*, 2013). ER morphology is complex and is subject to a constant remodeling in response to cellular demands. Cells that fail to support proper structural adaption and turnover of the ER are unable to cope with biological needs (Westrate *et al*, 2015; Nixon-Abell *et al*, 2016). Cells employ two major degradative pathways to maintain ER protein homeostasis: the ubiquitin–proteasome and the autophagy–lysosome system (Sun & Brodsky, 2019). Proteasomal degradation involves the retro-translocation of misfolded proteins and their cytosolic ubiquitination by the ER-associated protein degradation (ERAD) system (Ruggiano *et al*, 2014). Alternatively, degradation of defined portions of the ER, lipid membrane, and larger protein aggregates within or at the ER occurs via selective autophagy termed ER-phagy (Dikic, 2018; Wilkinson, 2020). Selectivity is driven by discerning autophagy receptors and their LIR domains, which bridge cargo and autophagosomal membranes (Birgisdottir *et al*, 2013; Stolz *et al*, 2014). To date, the mammalian ER-resident proteins FAM134B, SEC62, RTN3, CCPG1, ATL3, and TEX264 have been identified as ER-phagy receptors that mediate the coupling of ER fragments to autophagic membranes. These membrane proteins participate in basal ER turnover, ER reshaping following stress-related expansion, as well as lysosomal degradation of ER protein aggregates (Khaminets *et al*, 2015; Fumagalli *et al*, 2016; Grumati *et al*, 2017; Smith *et al*, 2018; An *et al*,

1  Telethon Institute of Genetics and Medicine (TIGEM), Pozzuoli, Italy
2  Institute of Biochemistry II (IBC2), Faculty of Medicine, Goethe University, Frankfurt am Main, Germany
3  Buchmann Institute for Molecular Life Sciences (BMLS), Goethe University, Frankfurt am Main, Germany
4  Department of Theoretical Biophysics, Max Planck Institute of Biophysics, Frankfurt am Main, Germany
5  Structural Genomics Consortium at BMLS, Goethe University, Frankfurt am Main, Germany
6  Institute of Human Genetics, Jena University Hospital, Friedrich-Schiller-University, Jena, Germany
7  Institute for Biophysics, Goethe University, Frankfurt am Main, Germany
   *Corresponding author. Tel: +39 081 1923 0688; E-mail: p.grumati@tigem.it
   **Corresponding author. Tel: +49 069 798 42589; E-mail: stolz@em.uni-frankfurt.de
   †These authors contributed equally to this work

2019; Chen *et al*, 2019; Chino *et al*, 2019). In addition, ER-phagy can also be driven by the two soluble autophagy receptors SQSTM1/p62 and CALCOCO1 in cooperation with the ER proteins TRIM13 and VAMPA/B, respectively (Ji *et al*, 2019; Nthiga *et al*, 2020). Besides classical macro-ER-phagy, two other forms of ER-phagy have been described in mammals: a piecemeal micro-ER-phagy, which occurs during recovery from cyclopiazonic acid-induced ER stress (Loi *et al*, 2019) and ER-derived single membrane vesicles, which deliver ATZ mutants to the lysosome for degradation (Fregno *et al*, 2018). Despite all the unknowns, we can be certain that the heterogeneity of the ER-phagy receptors confers ER-phagy with some levels of sub-specificity (Chino & Mizushima, 2020).

FAM134B, the first identified mammalian ER-phagy receptor to be identified, was shown to function under both basal and stress conditions (Khaminets *et al*, 2015). Initially, the physiological and pathophysiological role of FAM134B were described in the context of esophageal and colorectal cancers (Tang *et al*, 2007; Kasem *et al*, 2014) and in the pathogenesis of sensory and autonomic neuropathy (HSANII) (Kurth *et al*, 2009). Since then, FAM134B has been classified as an intra-membrane ER-resident protein that harbors a reticulon homology domain (RHD) and a LIR sequence (Khaminets *et al*, 2015; Bhaskara *et al*, 2019). Absence of FAM134B promotes ER expansion, which results in ER stress and subsequent neuropathies due to neuronal death. Conversely, overexpression of FAM134B induces a dramatic fragmentation and re-shuffling of ER membranes into autophagosomes (Khaminets *et al*, 2015). Transcriptional induction of *FAM134B* by the Mitf transcription factors TFEB and TFE3 is sufficient to induce ER-phagy (Cinque *et al*, 2020). Genetic inactivation of the FAM134B LIR domain abrogates ER fragmentation and subsequent lysosomal degradation (Khaminets *et al*, 2015). Moreover, misfolded pro-Collagen I was identified as a substrate for FAM134B-mediated ER-phagy (Forrester *et al*, 2019).

FAM134B belongs to the FAM134 protein family. The three FAM134 proteins are encoded by three different genes and share the same LIR amino acidic sequence (Khaminets *et al*, 2015). However, not much more is known about FAM134A and FAM134C. Here, we show that FAM134A and FAM134C are functional ER-phagy receptors with broad distribution throughout the ER network. They exhibit a limited ability to fragment ER under basal conditions, which can be augmented by environmental stresses. All three FAM134 proteins are on their own essential in maintaining ER morphology and protein homeostasis. However, while FAM134C seems to act in concert with and as a facilitator of FAM134B activity in the degradation of misfolded Collagen I, FAM134A drives a FAM134B- and LIR-independent degradation pathway. This study uncovers previously unstudied roles of FAM134A and FAM134C in governing ER shape and homeostasis.

## Results

### FAM134 localization, structure, and RHD dynamics

Looking at the predicted domain structures, all three FAM134 proteins carry a short N-terminal region and a reticulon homology domain (RHD), while the length of the C-terminus differs between paralogues. Despite the significant sequence divergence (< 30%

homology) between the three proteins, the LIR motif along with the flanking amino acids is highly conserved in all family members and across the species examined (Figs 1A and EV1A). Immunofluorescence microscopy of U2OS cells, expressing HA-tagged FAM134 proteins, highlighted the cellular localization of FAM134A and FAM134C. Both proteins were present throughout the ER network, as demonstrated by the overlap with the ER markers CALNEXIN (CANX) and REEP5 (Figs 1B and EV1B). On a more global scale, the three Fam134 proteins showed a broad distribution across organs and tissues. At least one of the Fam134 proteins was found in most of the analyzed murine organs and tissue samples (Fig EV1C). However, the protein levels were heterogeneous among the various tissues and Fam134 proteins were not always equally represented in the same tissues, suggesting tissue specific roles for each Fam134 paralogue. All three Fam134 proteins were abundant in the brain, liver, and heart while absent in pancreas and intestine (Fig EV1C). We detected major variabilities in lung, skeletal muscle, spleen and kidney where the Fam134 proteins were differentially expressed (Fig EV1C).

FAM134B binding to ER membranes is mediated by its RHD, which senses and actively induces membrane curvature and promotes ER-fragmentation during ER-phagy (Bhaskara *et al*, 2019). In order to investigate FAM134A and FAM134C dynamics, we built structural models of their RHDs. The RHD domain of FAM134 proteins forms wedge-shaped membrane inclusions in the ER membrane with four highly conserved structural elements: two ER-anchoring transmembrane helical hairpins (TM1,2 and TM3,4), that firmly anchor the RHD into the ER membrane, connected by a flexible cytoplasmic linker and two amphipathic helices ($AH_L$ and $AH_C$) that strongly interact with the cytoplasmic leaflet and flank the TM3,4 segment on both sides (Fig 1C and D). To further characterize the shape and dynamics of the three RHDs and their ability to perturbate the ER membrane, we built molecular models of the three RHDs and performed extensive coarse-grained molecular dynamics (CGMD) simulations in model bilayers. First, we simulated the behavior of individual RHD molecules embedded in POPC bilayers under periodic boundary conditions (up to 10 μs) (Movies EV1–EV3). We found that the main structural features of the RHD fragments (TM hairpins and AH-segments), including their membrane interactions, are well preserved among all the three FAM134 family members (Figs 1D and EV1D). Nevertheless, we observed differences, e.g., in charge distribution (Fig EV1E). Although all the three RHDs adopted a wedge-shape in the bilayer, we found that their shape and dynamics strongly differed (Figs 1E and EV1F). The RHD of FAM134A adopted a more compact shape throughout the simulation (Movie EV1). By contrast, the RHDs of FAM134B (Movie EV2) and FAM134C (Movie EV3) populated more open structures (Fig 1E). We reasoned that the differences in the distribution of the observed wedge shapes could be due to differences in the conformational dynamics of the individual RHDs. Therefore, we clustered the individual conformations sampled by the three different RHDs. Clustering of conformations sampled in the FAM134A-RHD trajectory resulted in a total of 3 clusters with a single dominant cluster (98.08%), explaining the narrow distribution of the radius of gyration. By contrast, a total of 11 and 10 clusters were observed for FAM134B-RHD and FAM134C-RHD, respectively. For FAM134B-RHD the clusters 1, 2, and 3 account for 69.3%, 14.4%, and 11.1% of the population, indicating that there are at least 3 major distinct

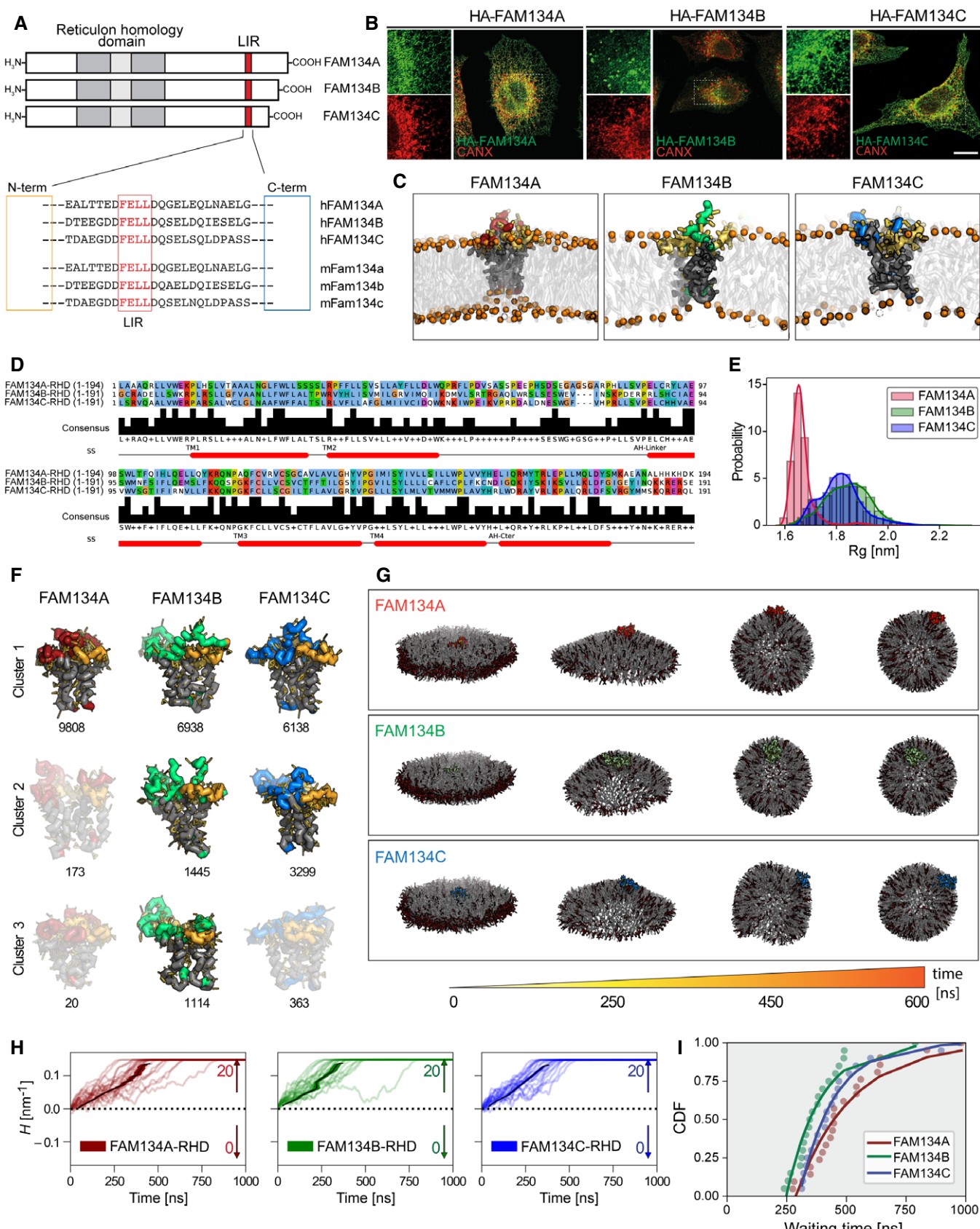

**Figure 1.**

**Figure 1. Structure and dynamics of human FAM134 family members.**

A  Schematic representation of FAM134 protein structures and sequence of the LC3-interacting region (LIR) domain of human (h) and mouse (m) proteins (red square).
B  Immunofluorescence images of U2OS cells expressing FLAG-HA-FAM134 after 24 h of doxycycline induction stained for HA (green) and endogenous calnexin (CANX; red). Scale bar: 20 μm.
C  Structural models of reticulon homology domains (RHDs) of FAM134 proteins were built and coarse-grained before embedding in POPC bilayers and simulated using MD simulations. The overall shape of the RHD along with its membrane footprint was monitored by measuring the radius of gyration $R_g$ of the protein (gray: TM segments; yellow: AH segments; red, green, blue: cytosolic loop of FAM134A, FAM134B, FAM134C).
D  Sequence alignment of human FAM134 proteins based on hydrophobicity profiles computed using AlignME. The helical structural elements characteristic of the RHD are denoted in red below along with a consensus sequence.
E  Probability density of $R_g$ for FAM134-RHDs, indicating compact (small $R_g$ value) or widespread (high $R_g$ value) conformations.
F  Distinct conformational states of the RHD of FAM134 proteins were obtained from conformational sampling (total $n = 10,000$) using coarse-grained molecular dynamic simulations (10 μs) and subsequent clustering. Representative conformations corresponding to the top 3 populated clusters are shown for each system. Conformation states in light colors represent the most populated states in the reported simulation.
G  Snapshots showing the curvature of bicelles containing DMPC (gray) and short-chain DHPC lipids (red) together with FAM134-RHDs over time.
H  Curvature time-traces (smoothed running averages over 11 ns widows) from individual replicates ($n = 20$) quantify the bicelle shape transformation process during simulations. Black lines (mean value) and shaded region (s.e.m) denote the average behavior of the system. From the 20 replicates, data on waiting times ($t$) were collected to compute rates and acceleration factors (Table EV3). Vesicle formation is marked by bilayer-disk curvature, IHI = 0.14 nm$^{-1}$.
I  Cumulative distribution functions (CDF) of recorded waiting times for formation of vesicle (filled circles: simulations; lines: Single Poisson process with additional lag time).

conformational states sampled. For FAM134C-RHD, we found two distinct conformations with 61.3% and 32.9% population, respectively (Fig 1F, Table EV1). We also found that fluctuations of FAM134A-RHD around the average structure were small, indicating a more rigid structure, whereas large fluctuations were observed in the exposed cytosolic loops for the RHDs of FAM134B and FAM134C, suggesting greater flexibility (Fig EV1G).

**FAM134A and FAM134C induce ER vesicles formation more slowly than FAM134B**

We performed *in silico* membrane curvature-induction assays to relate differences in shape and flexibility of the individual RHDs to their function in ER membrane fragmentation. Following (Bhaskara *et al*, 2019), we simulated discontinuous bicelle patches (DMPC + DHPC lipids) with embedded RHD molecules to study the kinetics of the curvature induction process. We found that all three RHDs could induce spontaneous bicelle-to-vesicle transitions within the simulation time (3 × 20 replicates; 1 μs each), confirming their intrinsic ability to actively induce membrane curvature (Fig 1G and H; Table EV2; Movies EV4–EV6). In all transitions, the membranes curved away from the cytoplasmic leaflet, as required for budding and subsequent fragmentation of the ER. By comparison, the RHDs of FAM134B and FAM134C appeared to induce bicelle-to-vesicle transitions more swiftly (Figs 1H and I, and EV1H; Movie EV5 and Movie EV6). We directly compared the driving force for RHD-induced curvature induction by measuring rates of vesicle formation and we estimate acceleration by factors of 176 and 206 for FAM134A-RHD and FAM134C-RHD, respectively. By contrast, FAM134B-RHD showed an acceleration in vesicle formation by a factor of 235 (Figs 1I and EV1I; Table EV3).

The more prominent and dynamic wedging of FAM134B-RHD is thus associated with faster vesicle formation, which would explain the high number of ER fragments observed for FAM134B in comparison with FAM134A and FAM134C (Fig 1B).

**FAM134A and FAM134C contain a classical and functional LIR motif**

The most conserved feature among the three FAM134 proteins is the LIR domain region, suggesting that this domain is important for

FAM134A and FAM134C activity. To test this hypothesis, we performed pull-down experiments with purified GST-mATG8s. In this experimental setting, FAM134A and FAM134C were able to bind to all six mATG8 proteins to different extent, while our controls Ub and tetra Ub remained unbound (Fig 2A). Consistently, mutation of the LIR motif in FAM134A and FAM134C or mutation of the classical binding site on LC3B was sufficient to abolish this interaction, thus supporting the presence of a functional, canonical LIR motif in FAM134A and FAM134C (Fig 2B and C). Similar results were obtained in mouse embryonic fibroblasts (MEFs), thus confirming the conservation of the LIR domain between human and mouse (Fig EV2A). Next, we sought to test whether FAM134A and FAM134C can act as bona fide autophagy receptors for the ER. It is well established that overexpression of wild-type FAM134B is sufficient to drive ER fragmentation (Khaminets *et al*, 2015). However, in agreement with our *in silico* data, overexpression of FAM134A rather resulted in a uniform distribution over the ER network, with few LC3B-positive ER fragments visible under basal conditions. Overexpression of FAM134C resulted in an intermediate phenotype (Figs 2 D and E, and EV2B and C). Upon nutrient starvation (EBSS), a strong increase in the number of ER fragments occurred in cells overexpressing FAM134A and FAM134C, respectively (Figs 2D and E, and EV2B and C). We aimed to distinguish between ER fragmentation prompted by starvation and the intrinsic impact of FAM134 overexpression. For this, the number of FAM134 (HA)-positive dots were quantified upon overexpression of LIR-mutant proteins, either after 2 h treatment with Bafilomycin A1 (Baf.A1) under basal or EBSS starvation conditions (Fig 2D and E, and EV2B and C). In contrast to the respective wild-type proteins, overexpression of FAM134A ΔLIR or FAM134C ΔLIR did not lead to an increased number of LC3B- and HA-positive ER fragments upon starvation (Figs 2D and E, and EV2B and C). Of note, the number of these structures increased only slightly upon EBSS treatment in FAM134B-overexpressing cells; however, they increased in size (Figs 2D and E, and EV2B and C). This suggests that FAM134B-mediated ER fragmentation function is fully active under basal conditions. Additionally, overexpression of FAM134B leads to significantly diminished ER branching under basal conditions, which did not drop further upon starvation. Genetic disruption of the FAM134B LIR domain abolished this effect (Figs 2F and EV2B and C) confirming that FAM134B-driven ER fragmentation relies on an intact LIR motif.

LIR-dependent activity of FAM134A and FAM134C did not significantly influence ER tubular structure (ER branching), neither under basal nor during starvation (Figs 2F and EV2B and C). Taken together, our results indicate that FAM134A and FAM134C can drive ER fragmentation in a LIR-dependent manner. Moreover, our data also indicate that the regulation of FAM134A and FAM134C, on a molecular level, differs from FAM134B. FAM134A and FAM134C appear to be relatively inactive under basal conditions and require an activation signal to fully engage.

## Overexpression of FAM134A and FAM134C promotes ER protein degradation

In order to be classified as an autophagy receptor a protein must simultaneously bind cargo and mATG8s, and be degraded together with the cargo (Birgisdottir et al, 2013; Stolz et al, 2014). All three FAM134 proteins can promote ER fragmentation and bind mATG8s (Fig 2). To further classify them as ER-phagy receptors, we demonstrated their ability to deliver discrete portions of the ER to the lysosome in a LIR-dependent manner. Immuno-gold labeling in U2OS cells overexpressing FAM134 highlighted the presence of all three FAM134 along with discrete portions of ER inside autophagosomes (Fig EV2D). We further confirmed the presence of FAM134 proteins inside the lysosomal compartment by co-staining for HA-FAM134 and endogenous LAMP1: wild-type FAM134s were detected inside LAMP1-labeled structures, while LIR mutant proteins were not (Fig EV2E). FAM134A and FAM134C protein levels dropped upon starvation, indicating that they are degraded along with their cargo (Fig EV2F and G). Consistently, upon starvation, LAMP1-positive structures were enriched with CANX-positive ER fragments that also colocalized with HA-FAM134s (Fig EV3). To determine whether ER degradation is directly mediated by FAM134 proteins, we performed full proteome mass spectrometry (MS) analysis on different cell lines under basal or starvation (8 h EBSS) conditions (Fig 3A). We then took the proteome of U2OS cells, expressing only endogenous levels of FAM134, and compared them to U2OS cell lines overexpressing individual FAM134 proteins under a doxycycline-controlled promotor. Principal component analysis (PCA) revealed expected clustering among replicates (Fig 3B). The induction of FAM134A and FAM134C leads to mild changes in the global proteome and respective datasets clustered closely to doxycycline treated control cells and non-induced (−Dox) cell lines (Fig 3B and C). The clustering distance from control cells increased upon EBSS treatment, leading to well-separated clusters. These findings support our previous observation that FAM134A and FAM134C are relatively inactive under basal conditions and require a stressor to be fully activated. As expected, overexpression (+Dox) of FAM134B alone was sufficient to cause a clear shift in the global proteome and EBSS treatment further enhanced this effect (Fig 3B; Dataset EV1). Notably, cells overexpressing FAM134B clustered separately from FAM134A and FAM134C overexpressing cells under the various experimental settings (Fig 3B; Dataset EV1). Significantly altered proteins were determined by an ANOVA test (FDR < 0.01) and reported in a heatmap (Fig 3C; Dataset EV1). Doxycycline-dependent expression of HA-FAM134 marked a shared cluster of significantly downregulated proteins related to the endoplasmic reticulum. FAM134-dependent downregulation of this cluster got even more prominent upon starvation (Fig 3C–F; Dataset EV1 and Dataset EV2).

The cluster of 149 proteins downregulated in a FAM134-dependent manner may contain shared ER-phagy substrates of the FAM134 family.

## FAM134A and FAM134C protein levels influence ER-phagy flux

ER-phagy is a dynamic process; therefore, we aimed to measure how the three FAM134 proteins singularly influence the ER-phagy flux dynamics over-time. To this end, U2OS cells overexpressing individual FAM134 proteins were complemented with a constitutively expressed ssRFP-GFP-KDEL ER-phagy reporter (Chino et al, 2019). Since GFP, but not RFP, is rapidly quenched by the acidic environment of the lysosome, ER-phagy flux can be discerned by the RFP/GFP ratio calculated based on microscopic images (Fig 3G). In our assay, DMSO served as the vehicle control, while Torin1 was used to activate autophagy (Fig 3H). Comparing the ratio of basal ER-phagy flux (DMSO) of the parental U2OS line and cells overexpressing individual FAM134 proteins, we recorded a mild induction of basal flux following the overexpression of each of the three FAM134 family members. At first glance, the similar levels of autophagy flux between all three FAM134-overexpressing cell lines seemed to contradict our observation that FAM134B is a potent inducer of ER fragmentation (Fig 2D). However, we noticed that only a fraction of the FAM134B-positive ER-fragments were decorated with LAMP1 under basal conditions (Fig EV2E) and therefore ascertainable by our ER-phagy flux assay.

Delivery of the remaining, LAMP1-negative, FAM134B-positive ER-fragments to lysosome apparently exceeds the capacity of the cellular transport system. Torin1 treatment enhanced ER-phagy flux over time and FAM134 overexpression amplified the effect, indicating that the expression level of FAM134 proteins is a limiting factor of ER-phagy under this condition (Fig 3H). We also analyzed changes in the protein levels of two ER-resident proteins, SEC22B and REEP5. Protein levels of SEC22B and REEP5 dropped upon overexpression of FAM134 proteins (Fig 3I), thereby validating our ER-phagy flux data. To induce ER specific stress, we employed Thapsigargin (Tg), a non-competitive inhibitor of the sarco/endoplasmic reticulum Ca$^{2+}$ ATPase (SERCA). SERCA inhibition leads to the depletion of ER calcium and subsequently to the activation of the unfolded protein response (UPR; Sehgal et al, 2017). We aimed to find a concentration, which would cause only mild ER stress triggering ER-phagy and not autophagy arrest and subsequent cell death (Ganley et al, 2011; Lindner et al, 2020; Xu et al, 2020). We therefore measured ER-phagy flux in the presence of different concentrations of Tg (50 nM to 2 µM). Thapsigargin treatment led to greater ER-phagy flux in FAM134 overexpressing cells when compared to the parental cell line in a dose-dependent manner (Fig 3J). Concentrations exceeding 200 nM Tg had a negative effect on the maximal induction of ER-phagy flux in all tested cell lines (Fig 3J).

Considering that overexpression of FAM134 proteins enhances ER-phagy flux, we wondered if their absence would have the opposite effect. To test this, we isolated mouse embryonic fibroblasts (MEFs) from Fam134a, Fam134b, and Fam134c single knockout mice, reconstituted them with either the respective wild-type or ΔLIR mutant Fam134 protein, and complemented the cells with the ssRFP-GFP-KDEL reporter (Fig 4A and B). Of note, we did not observe a compensatory increase in protein levels of the remaining

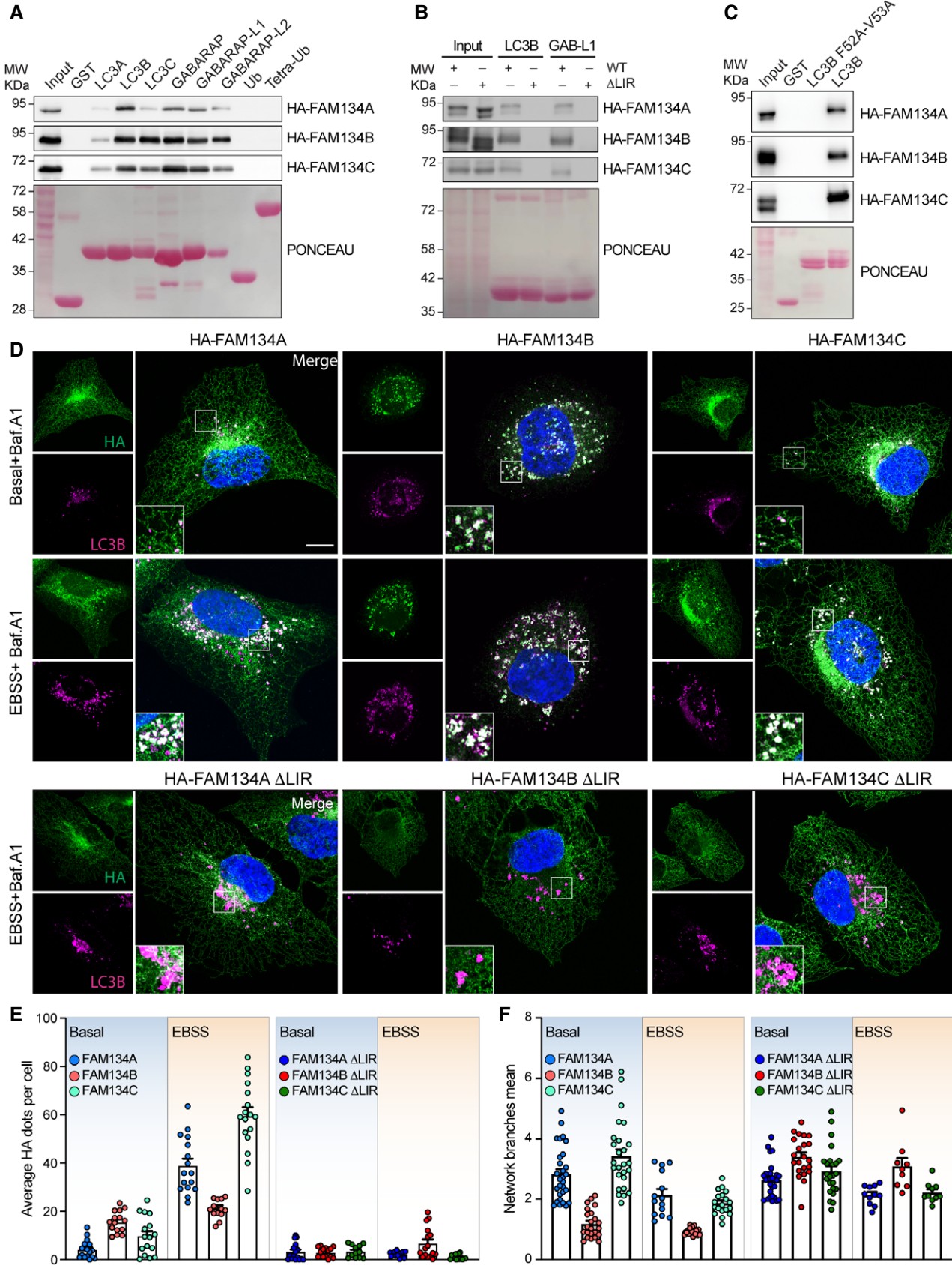

**Figure 2.**

**Figure 2. FAM134 members are ER resident proteins harboring a functional LIR domain.**

A–C Lysates of HEK293T cells transiently transfected with HA-tagged wild-type (WT, Fig 2A–C) and wild-type or ΔLIR mutant (Fig 2B) *FAM134* genes were subjected to pull-down experiments (representative data; WB detection against HA; *n* = 3) using A, B) purified GST, wild type GST-LC3s, GST-GABARAPs, GST-Ubiquitin (Ub), GST-Tetra-Ubiquitin, or C) GST-LC3B and GST-LC3B F52A-V53A as baits. A representative Ponceau staining is shown.

D Immunofluorescence images of U2OS expressing wild-type or ΔLIR mutant FLAG-HA-FAM134 proteins after 24 h of doxycycline induction under different conditions (basal = DMSO, Baf.A1 = 2 h 200 nM Bafilomycin A1, EBSS = 2 h starvation in EBSS) Scale bar: 10 μm; staining against FAM134 (HA; green) and endogenous LC3B (red).

E Automatic quantification of HA-positive dot-like structures. Each data point represents the average of dots per cell of one view (representative image see Fig EV2B). Total number of cells analyzed for basal: FAM134A/B/C/AΔLIR/BΔLIR/CΔLIR = 592/472/358/280/233/300; EBSS: FAM134A/B/C/AΔLIR/BΔLIR/CΔLIR = 364/720/142/364/720/674; error bars indicate s.e.m.; biological replicates WT/ΔLIR = 3/1.

F Automated quantification of ER branching of in an automated manner (Valente *et al*, 2017). Each datapoint represents one view (representative image see Fig EV2B). Same cells as in (E); error bars indicate s.e.m.

Source data are available online for this figure.

two Fam134 paralogues in single knockout MEFs (Fig 4A). Under basal conditions (DMSO), ER-phagy flux only marginally differed between knockout and reconstituted cells. In contrast, upon Torin1 treatment all three knockout cell lines reconstituted with their respective wild-type Fam134 proteins showed higher ER-phagy flux in comparison with their parental knockout cell lines (Fig 4C). The same effect was observed when treating cells with different concentrations of Tg (Fig 4D and E). These results are in line with our data obtained in U2OS cells (Fig 3H and I) and suggest that presence of sufficient levels of Fam134 proteins is important for the regulation of ER turnover via ER-phagy. Of note, reconstitution with the Fam134b ΔLIR mutant negatively affected ER-phagy flux (Fig 4C–E).

## Fam134 family members have unique and shared functions

Loss of ER-phagy receptors often negatively impacts the clearance of their substrates (Schultz *et al*, 2018; Cunningham *et al*, 2019; Forrester *et al*, 2019). To investigate the role of murine Fam134a and Fam134c in ER protein homeostasis, we performed MS analyses comparing the global proteome of wild-type and knockout MEFs (Fig 5A). Principal component analysis (PCA) and heat-mapping of more than 4,800 proteins revealed that proteomic profiling can efficiently discriminate samples of different genotypes (Figs 5B and EV4A and B; Dataset EV3). This indicates that loss of a single Fam134 protein causes profound changes in the global proteome. The drivers of the discrimination between wild-type and *Fam134* knockout MEFs were significantly enriched for proteins annotated to endoplasmic reticulum and Collagen type I (Fig 5C; Dataset EV3). At the same time, the proteome of *Fam134b* and *Fam134c* knockout MEFs clustered closer and with distance to the proteome of *Fam134a* knockout MEFs (Fig 5B), indicating a higher functional overlap between Fam134b and Fam134c. Proteins with a difference > 2fold ($Log_2 > 1$) and a $-Log_{10}$ *P*-value > 1.3 were counted as significantly different between *Fam134* knockout and wild-type MEFs (Fig EV4C; Dataset EV3). Venn diagrams of significantly up- and down-regulated proteins, in *Fam134* knockout versus wild-type MEFs, indicated an intricate relationship among the three genotypes (Fig EV4D). By performing an unsupervised clustering of ANOVA-significant proteins, we obtained nine separate clusters sorted along their GO cellular compartment assignments (Figs 5D and EV4E; Dataset EV3 and Dataset EV4). We focused on cluster 8 comprised of 197 proteins linked to the ER. Fitting to our hypothesis that Fam134 proteins are important for

efficient ER-turnover, the averaged protein level of the cluster was increased in all *Fam134* knockout cells (Fig 5D). The 197 proteins are part of a protein–protein interaction network, which allowed us to discern for three highly inter-connected protein clusters encompassing endoplasmic reticulum proteins, Collagens and collagen-processing enzymes, and stress fiber components (Figs 5D–F and EV4F and G). One of the proteins enriched in all three *Fam134* knockout cell lines was Climp63, which we further validated using fluorescent microscopy (Fig 5G and H). Of note, this is exactly the opposite of what we found in U2OS cells, where the overexpression of FAM134 proteins led to a drop of several ER protein levels (Dataset EV1).

Accumulation of Collagens, particularly Collagen I, was one of the most striking alterations in the global proteome of all three *Fam134* knockout cells (Figs 5F and I, and EV4C and G). A dysfunction in misfolded pro-Collagen I degradation was previously reported for *Fam134b* knockout cells (Forrester *et al*, 2019). Immunolabeling for Collagen I confirmed that it massively accumulates in the cytosol of *Fam134a, Fam134b,* and *Fam134c* knockout MEFs (Fig 5J). These results indicate that, at least in MEFs, all three Fam134 family members have equally important roles in maintaining a proper pro-Collagen I quality control. Collagen I continued to be regularly secreted in *Fam134* knockout cells (Fig EV4H) indicating that absence of Fam134 proteins does not interfere with the secretory pathway itself. Of note, *Fam134b* and *Fam134c* knockout MEFs accumulated more Collagen I compared to *Fam134a* knockout cells (Fig EV4C and H). Our findings indicate that the absence of Fam134 proteins impaired Collagen homeostasis and protein quality control through reduced degradation of its mis-folded forms.

## Fam134a and Fam134c are essential for maintaining ER homeostasis

To further characterize the role of Fam134 family members in ER homeostasis, we subjected *Fam134* single knockout MEFs to electron microscopy analysis. Compared to wild-type cells, the ER in *Fam134a* and *Fam134c* knockout cells appeared swollen and dilated (Fig 6A), indicating a significantly disturbed ER morphology. This phenotype was also found in *Fam134b* knockout cells (Fig 6A) (Khaminets *et al*, 2015). Immunofluorescence staining confirmed that the swelling of the ER corresponded to expanded areas containing the ER markers Climp63 and Canx (Fig 6B). In order to determine whether the function of Fam134a and Fam134c in ER shaping was LIR-dependent, knockout MEFs were reconstituted with either wild-

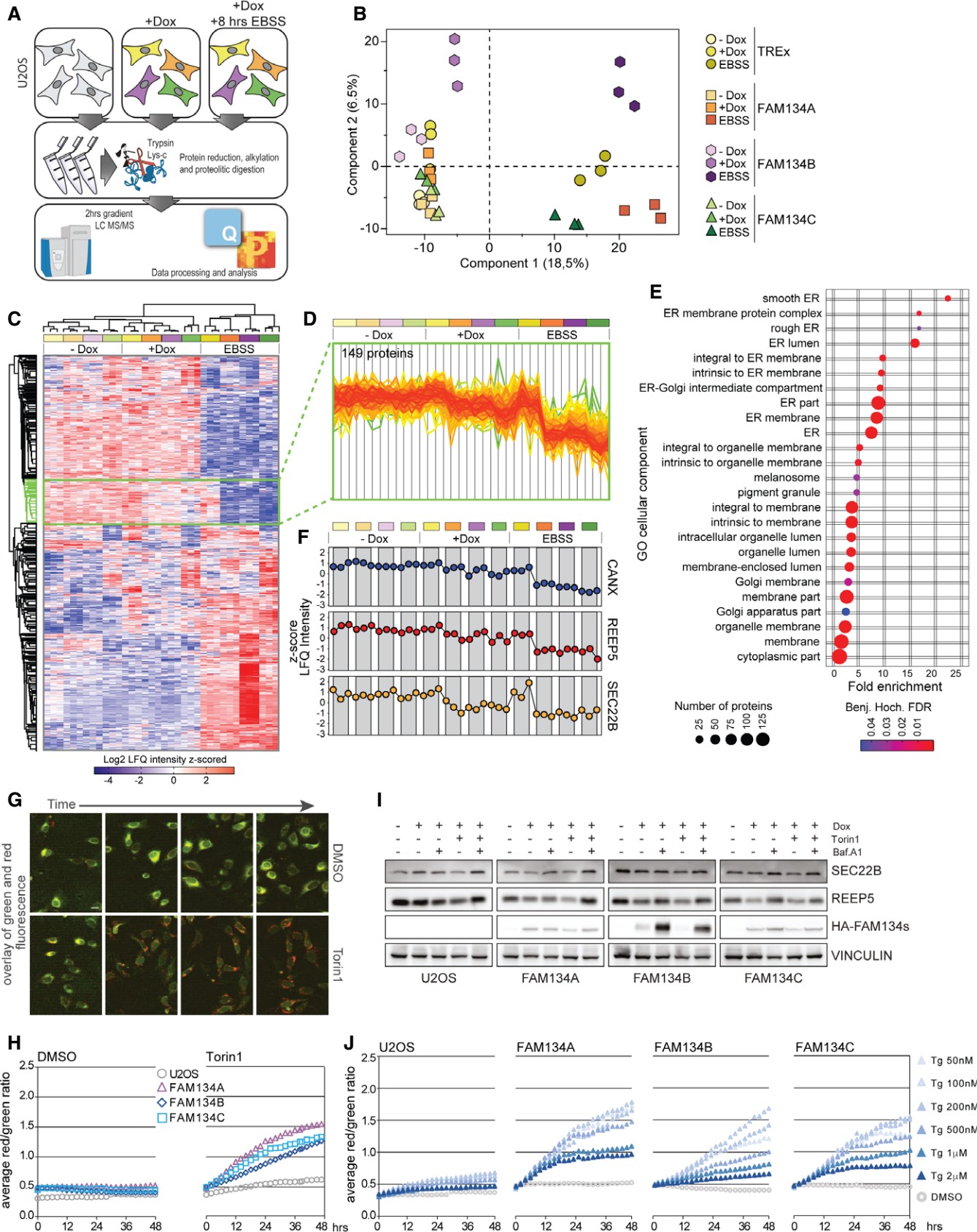

Figure 3.

**Figure 3.  FAM134s overexpression promotes the degradation of ER proteins.**

A   Schematic representation of the experimental procedure to obtain the proteomes of U2OS control cells (−Dox) or cells inducible expressing FLAG-HA-FAM134 proteins upon 24 h of doxycycline induction (+Dox) by mass spectrometry (MS). Cells were grown in basal condition or starved with EBSS for 8 h prior to lysis.
B   Principal Component Analysis of the MS data after indicated treatments.
C   Hierarchical clustering of label-free quantitation (LFQ) intensities of significantly changed proteins (ANOVA, FDR 0.01). Cluster framed in green includes proteins with a "endoplasmic reticulum" enriched GO.
D   Profile-plot of protein levels (LFQ intensities) of the cluster enriched for "endoplasmic reticulum" upon indicated treatments.
E   List and frequency of GOCC terms related to the protein cluster in (D).
F   CANX, REEP5 and SEC22B as sample proteins extracted from (D).
G   Small sections of sample pictures of U2OS cells stably expressing the ER-phagy reporter ssRFP-GFP-KDEL. Pictures were obtained by the IncuCyte® S3 (10×) and show the overlaid GFP and RFP signal from DMSO and Torin1-treated cells over time. Scale bar 25 μm. Time lapse of representative total image views can be found in Movies EV7 and EV8.
H   ER-phagy flux, represented by the RFP/GFP ratio (total fluorescent intensities), upon basal (DMSO) or autophagy induced (Torin1) conditions, in U2OS control cells and cells overexpressing FAM134 proteins upon 24 h of Dox induction.
I   Representative Western blot of total cell lysates of U2OS cells overexpressing indicated FAM134 proteins upon 24 h of Dox induction. Autophagy flux was induced and blocked by treatment with Torin1 (8 h) and Bafilomycin A1 (Baf.A1, 8 h), respectively. $n = 3$.
J   Comparison of DMSO- or Thapsigargin (Tg)-treated U2OS ssRFP-GFP-KDEL cell lines overexpressing indicated FAM134 family members upon 24 h Dox induction.

Data information: all ER-phagy flux data are representative of three biological replicates and presents averaged data obtained from three individual wells (technical replicates) via the IncuCyte® S3, each view containing > 150 cells.
Source data are available online for this figure.

type or the LIR mutant of the respective Fam134 protein (Fig 6C). Reconstitution of knockout MEFs with the wild-type protein resulted in a reduction of Climp63 and Canx-positive areas to the level of wild-type cells (Fig 6B and D). In contrast, cells reconstituted with Fam134-ΔLIR contained Climp63 and Canx positive areas that were equal or even greater than observed in knockout cell lines. This suggests that the role of Fam134s in maintaining ER shape is LIR-dependent (Fig 6B and D) and that expression of Fam134bΔLIR seems to have a dominant negative effect on ER degradation (Figs 6D and 4C–E).

### Fam134a mediates pro-Collagen I degradation in a unique, LIR-independent manner

In wild-type MEFs, large amounts of misfolded pro-Collagen I were found in Lamp1 positive lysosomes, while these events were rare or altogether absent in all *Fam134* knockout cell lines (Fig EV5A). As expected, reconstitution of the knockout MEFs with the respective wild-type protein diminished accumulation of pro-Collagen I (Figs 7A–C, and EV5A). Additionally, we found FAM134A and FAM134C to interact with CANX (Fig EV5B–D). CANX may function as a co-receptor by recognizing misfolded pro-Collagen I in the ER lumen and simultaneously interacting with the ER-phagy receptor (Forrester *et al*, 2019). Surprisingly, reconstitution of *Fam134a* knockout MEFs with the ΔLIR mutant of Fam134a was equally efficient at counteracting the accumulation of pro-Collagen I as the wild-type protein. This was not the case for the reconstitutions with Fam134b and Fam134c ΔLIR mutants (Figs 7A–C and EV5A). The similar phenotypes of individual *Fam134* knockout MEFs may suggest that all three Fam134 family members act on the same pathway, either sequentially or simultaneously in an oligomeric complex. However, the distinct clustering of the *Fam134a* knockout proteome (Fig 5B) made us wonder, if Fam134a may act in a parallel, Fam134b-independent pathway. To test this hypothesis, we used our *Fam134a* constructs to significantly increase the protein levels of Fam134a in *Fam134b* knockout cells (Fig EV5E). Indeed, overexpression of wild-type or ΔLIR mutant Fam134a was able to rescue the pro-Collagen-I accumulation in *Fam134b* knockout MEFs (Fig 7

D and E). Knowing that FAM134A is in principle capable of binding LC3B (Fig 2A), we tested the interaction between FAM134A with LC3B in cells (Fig EV5F). In agreement with the fact that Fam134aΔLIR compensated for the loss of Fam134b, we were unable to detect a strong interaction between FAM134A and endogenous LC3B. However, we did note a slight interaction between FAM134A and the GABARAP proteins (Fig EV5F and G). In contrast to Fam134a, overexpression of Fam134c did not compensate for the loss of Fam134b (Figs 7F and G, and EV5H), indicating that Fam134c may either cooperate or act in complex with Fam134b to deliver misfolded pro-Collagen I toward the lysosome. Notably, excess of misfolded pro-Collagen I phenotype in *Fam134c* knockout cells can be rescued by overexpression of both Fam134a and Fam134b; however, Fam134a overexpression had a stronger effect compared to that of Fam134b (Figs 7H and I, and EV5I). Finally, overexpression of either Fam134b or Fam134c in *Fam134a* knockout MEFs rescued the accumulation of misfolded pro-Collagen I (Fig 7J–M). These findings support the hypothesis that Fam134b and Fam134c may act on the same ER-phagy route, while Fam134a appears to function in parallel and in a Fam134b-independent manner.

## Discussion

In this study, we investigated the functional roles of FAM134 protein family members in regulating ER-phagy and ER remodeling pathways. We demonstrated that FAM134A/RETREG2 and FAM134C/RETREG3 carry a functional LIR motif, localize to the ER, and participate in the regulation of ER-phagy. Differences in the structure and dynamics of their reticulon homology domains (RHDs) as well as in the global proteomics of overexpressing and knockout cells point to some distinct and some shared functions between the two paralogues as well as their well-characterized relative FAM134B/RETREG1, which was the first ER-phagy receptor identified in mammalian cells (Khaminets *et al*, 2015).

Thus far, six membrane ER-specific autophagy receptors have been characterized (Chino & Mizushima, 2020) and, as this study

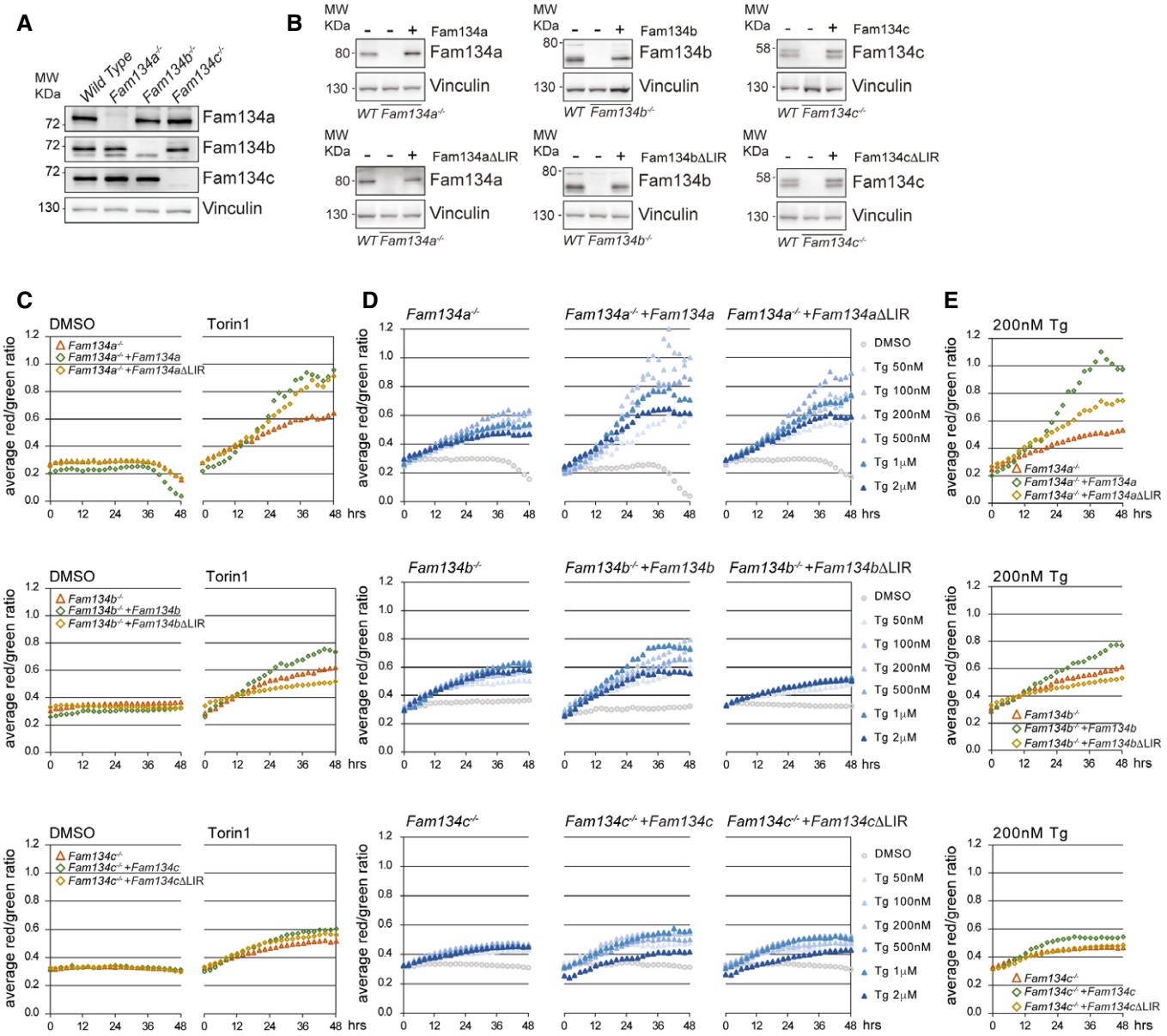

**Figure 4. FAM134 family members induce ER-phagy flux.**

A    Representative Western blot for Fam134 proteins in single knockout mouse embryonic fibroblasts (MEFs).
B    Western blot for Fam134 proteins in indicated MEFs, expressing the ER-phagy reporter ssRFP-GFP-KDEL and being reconstituted with the respective wild-type or ΔLIR-mutant Fam134 protein.
C–E    ER-phagy flux in ssRFP-GFP-KDEL *Fam134* knockout and reconstituted MEFs, represented by the RFP/GFP ratio (total fluorescent intensities). Data are representative for three biological replicates and present averaged data obtained from three individual wells (technical replicates) via the IncuCyte® S3, each view containing > 150 cells. (C) ER-phagy flux in basal-induced (DMSO) or autophagy-induced (Torin1) conditions. (D, E) ER-phagy flux in Thapsigargin (Tg)-treated MEFs.

Source data are available online for this figure.

suggests, there are likely many more yet to be discovered. The reason behind this number is the need of cells to respond and adapt to a variety of environmental changes. At first sight, the impact of FAM134A and FAM134C on the ER seems negligible: overexpression neither impacts ER branching nor does it lead to extensive fragmentation. However, the presence of an additional, autophagy-inducing trigger (Torin1, starvation, Thapsigargin-induced ER stress) seems to provide an activation signal for FAM134A and FAM134C. Under these conditions, both proteins actively drive ER-phagy and the amount of FAM134A and FAM134C seems to be a limiting factor in this process. As such, overexpression of either of the two paralogues under conditions, where ER-phagy is switched

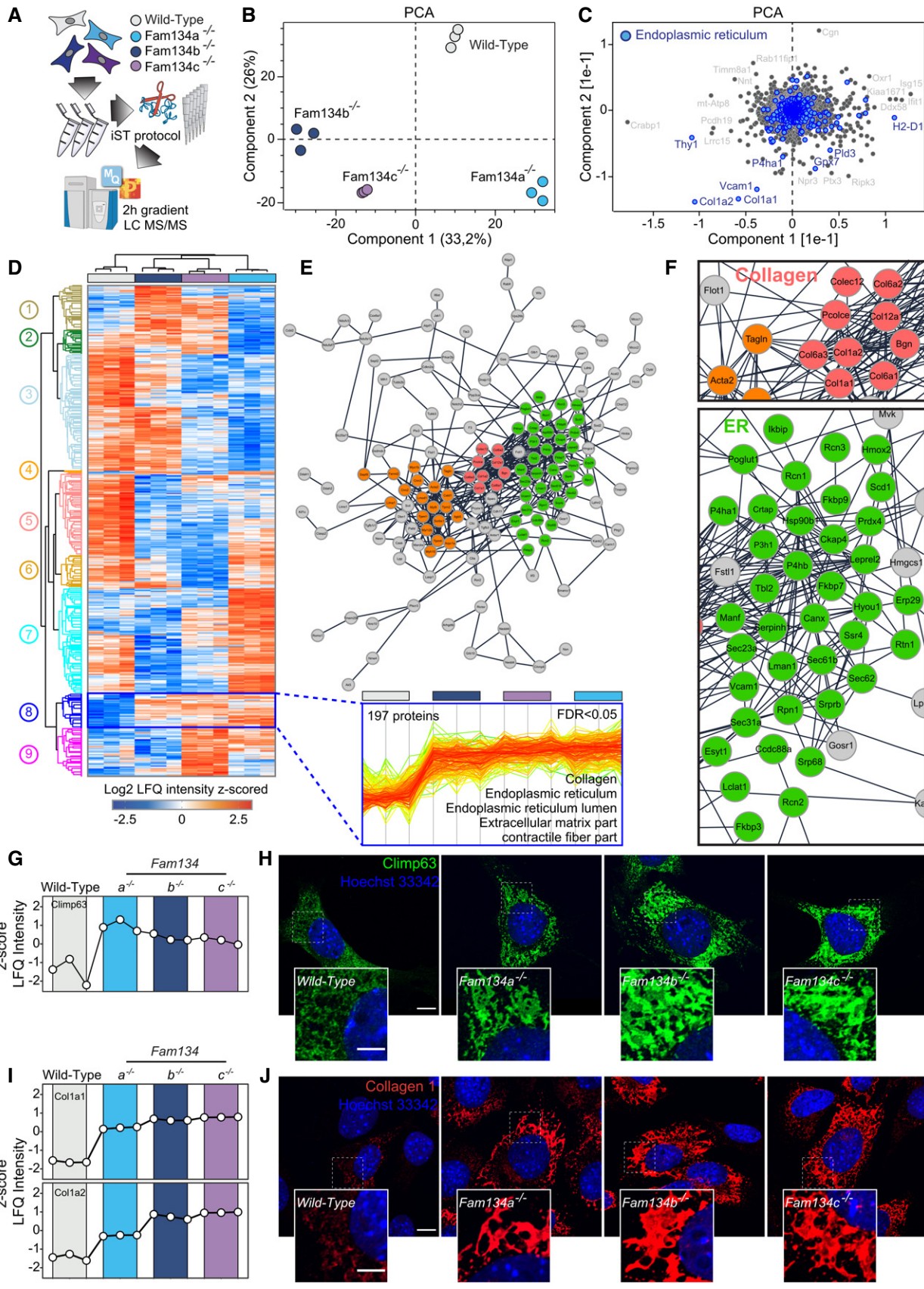

Figure 5.

Figure 5. Global proteome analysis of Fam134s knockout MEFs.

A   Schematic representation of the experimental procedure to obtain the proteomes of wild-type and *Fam134s* knockout MEFs by mass spectrometry (MS).
B   Principal component analysis (PCA) of analyzed proteomes.
C   Scatter plot showing proteins responsible for driving sample segregation along component 1 and component 2. Proteins driving segregation are significantly enriched (FDR < 0.05) for GO cellular component (GOCC) terms associated with endoplasmic reticulum (ER) (blue dots).
D   Unsupervised clustering of ANOVA-significant proteins by GOCC terms. Inset showing the profile plot and the GO-term enrichment (FDR < 0.05) of 197 proteins belonging to the highlighted cluster.
E   Physical interaction network of proteins significantly upregulated in *Fam134* knockout MEFs. The network was generated using the STRING plugin in the Cytoscape environment with a confidence score set to 0.4. Nodes are protein entities while lines represent physical interactions Red, orange, green: Collagen, stress fiber, ER associated.
F   Insets of (E) showing the Collagen and the ER cluster, respectively.
G, H   Profile plots showing the $Log_2$ LFQ intensity (G) and representative immunofluorescence (H) of Climp63 (green) in wild-type and *Fam134* knockout MEFs.
I, J   Profile plots showing the $Log_2$ LFQ intensity (I) and representative immunofluorescence (J) for collagen I (red) expression in wild-type and *Fam134* knockout MEFs. Scale bar: 10 μm. Inset scale bar: 5 μm. Nuclei were stained with Hoechst 33342.

on, leads to a strong increase of the maximum ER-phagy flux. How this activation takes place on a molecular level is currently unknown but may be mediated by post-translational modifications or changes in the interactome. Whether FAM134A and FAM134C are in addition regulated at the transcriptional level, as recently reported for FAM134B (Kohno *et al*, 2019; Cinque *et al*, 2020), remains as well to be investigated.

For FAM134B, which is fully active under basal conditions, the RHD and a flanking membrane-associated helix ($AH_C$) have been identified as drivers of membrane curvature induction, regulators of vesicle size, and ER fragmentation (Bhaskara *et al*, 2019). A phosphorylation site within the FAM134B-RHD was reported to enhance the protein oligomerization and its ability to fragment ER membrane (Jiang *et al*, 2020). The two TM hairpins within the RHD adopt closed and open conformations with different degrees of wedging (Bhaskara *et al*, 2019). The asymmetric shape of the RHD favors localization in highly curved ER regions, which further promotes FAM134B clustering and facilitates ER fragmentation by budding (Khaminets *et al*, 2015; Bhaskara *et al*, 2019; Siggel *et al*, 2021). The RHDs of FAM134A and FAM134C share key structural features with FAM134B-RHD, suggesting that they also engage in ER fragmentation. However, in line with experimental data, our *in silico* studies on FAM134A and FAM134C revealed differences in the composition and structural dynamics of their RHD domains. While FAM134B-RHD was highly dynamic, our modeling revealed only one and two main conformational states for FAM134A-RHD and FAM134C-RHD, respectively. As a consequence of the reduced dynamics, these proteins also showed slower rates for vesicle formation from bicelles. The lower calculated acceleration factors for vesicle formation are consistent with the limited ER-fragmentation ability of FAM134A and FAM134C in cells under basal conditions. It will be interesting to study in future, if the dynamics of RHDs of FAM134A and FAM134C can be as well influenced and enhanced by PTM.

It is important to note that Fam134a and Fam134c also provide critical functions in maintaining ER homeostasis under basal conditions and in absence of additional stresses. Massive deformation of the ER network and ER swelling in *Fam134a* and *Fam134c* knockout MEFs, point toward an essential function of these two proteins in maintaining ER fitness. Reconstitution with the respective wild-type proteins, but not with the LIR mutants, can revert this phenotype. Since knockout of all three *Fam134* genes have a similar ER swelling phenotype, we got interested in differences on the proteome level. We found the global proteome of MEFs depleted for individual *Fam134* genes to be very different from wild-type MEFs one. We clustered significantly altered proteins via their GO terms and found proteins within these clusters to follow quite distinct patterns of increased/decreased protein level upon knockout of individual *Fam134* genes. The cluster containing ER related proteins was commonly upregulated in all *Fam134* knockout MEFs. The opposite was true for the proteome of U2OS cells overexpressing individual FAM134 paralogues: here, a comparable cluster of ER-related proteins was commonly downregulated. In-depth analysis of all protein clusters may reveal distinct ER-phagy pathways regulated positively or negatively by individual FAM134 family members.

Collagen I was within the cluster of proteins accumulated in all three *Fam134* knockout MEFs. So far, only Fam134b has been characterized as a key mediator of pro-collagen I degradation via the lysosome (Forrester *et al*, 2019). Our findings indicate that Fam134a and Fam134c are equally important for maintaining proper pro-Collagen I homeostasis. We therefore used pro-Collagen I as a proof-of-concept substrate to investigate the interplay between Fam134 paralogues. The fact that Collagen I was still processed and secreted in all three *Fam134* knockout cell lines—proportionally to its total protein level—argues for an intact secretory pathway in the absence of individual Fam134 proteins. The increase in pro-

Figure 6. FAM134 absence causes ER expansion.

A   Electron microscopy of wild-type and *Fam134* single knockout MEFs (ER = endoplasmic reticulum). Scale bar: 500 nm.
B   Immunofluorescence images stained for endogenous Calnexin (Canx) or Climp63 in wild-type (WT), *Fam134* knockout MEFs, and *Fam134* knockout MEFs reconstituted with the respective wild-type or ΔLIR mutant Fam134 protein. Scale bar: 10 μm. Inset scale bar: 5 μm.
C   Representative Western blots of total cell lysate derived from MEFs used in (B) and detected for endogenous Fam134 and Actin, respectively.
D   Quantification of (B). At least $n = 10$ independent cells across three independent experiments each condition. The statistical significance was estimated by unpaired Student's *t* test. All data are represented as mean ± s.e.m. and the statistical significance is defined as *$P < 0.05$, **$P < 0.01$, ***$P < 0.001$.

Source data are available online for this figure.

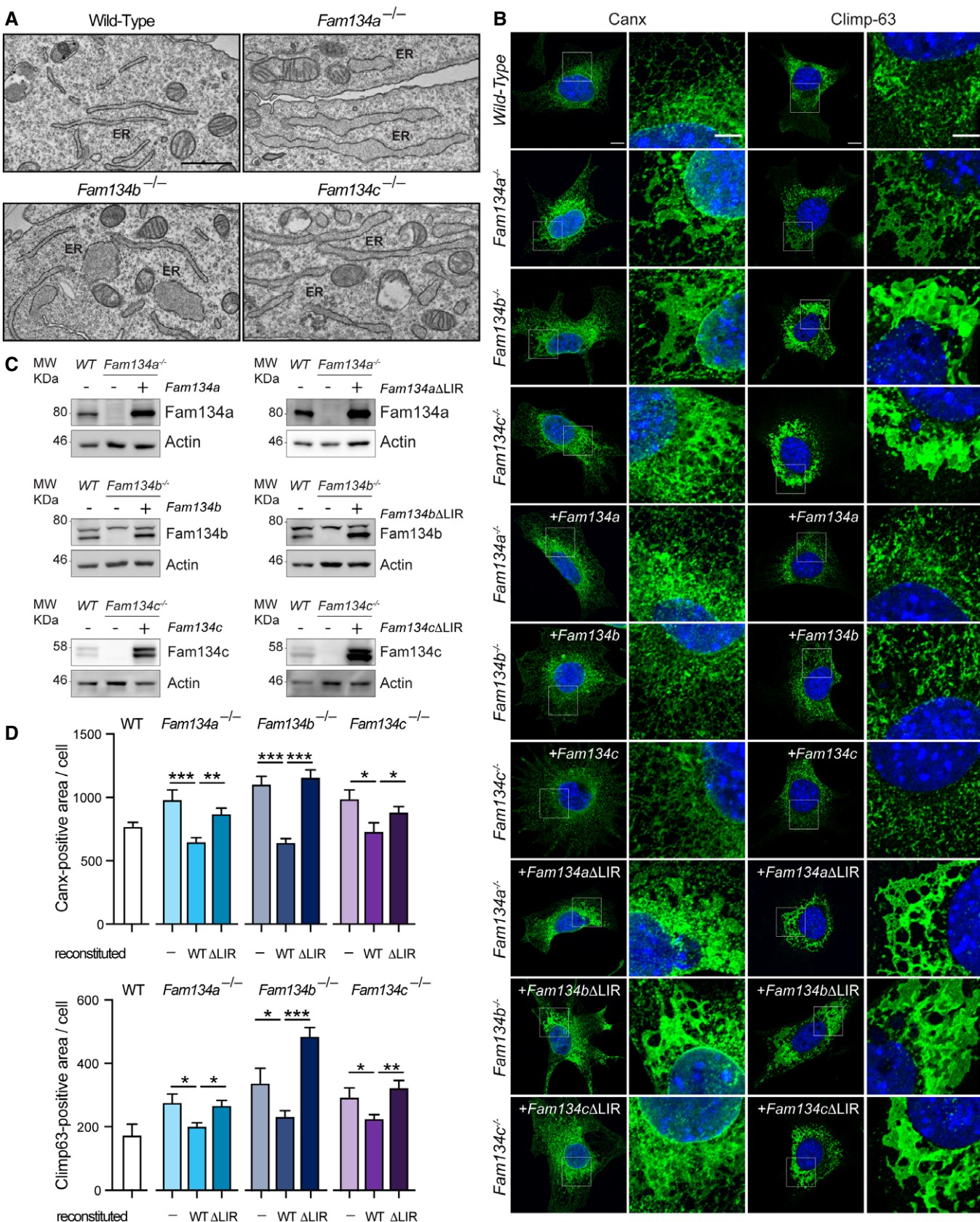

**Figure 6.**

Collagen I levels in the knockout cell lines is therefore a result of reduced degradation of the misfolded pro-Collagen I fraction rather than an impairment in its secretion. Furthermore, we observed that the overexpression of Fam134a in *Fam134b* or *Fam134c* knockout MEFs is sufficient to reduce pro-Collagen I to wild-type levels. In contrast, Fam134c does not seem to act on its own, because overexpression of Fam134c in *Fam134a* knockout MEFs only partially reverted the high Collagen I levels and did not rescue the Collagen I accumulation in *Fam134b* knockout MEFs. We deduce that Fam134c

may act as a co-receptor or enhancer in Fam134b-dependent ER-phagy of misfolded pro-Collagen I. This hypothesis would explain why overexpression of Fam134c does not diminish Collagen I accumulation in the absence of Fam134b, while overexpression of Fam134b in the absence of Fam134c does.

It is intriguing that the LIR motif of Fam134a appears to be dispensable for Collagen I degradation, but essential for Fam134a-dependent maintenance of proper ER morphology. The weak interaction between FAM134A and LC3B in U2OS cells would suggest

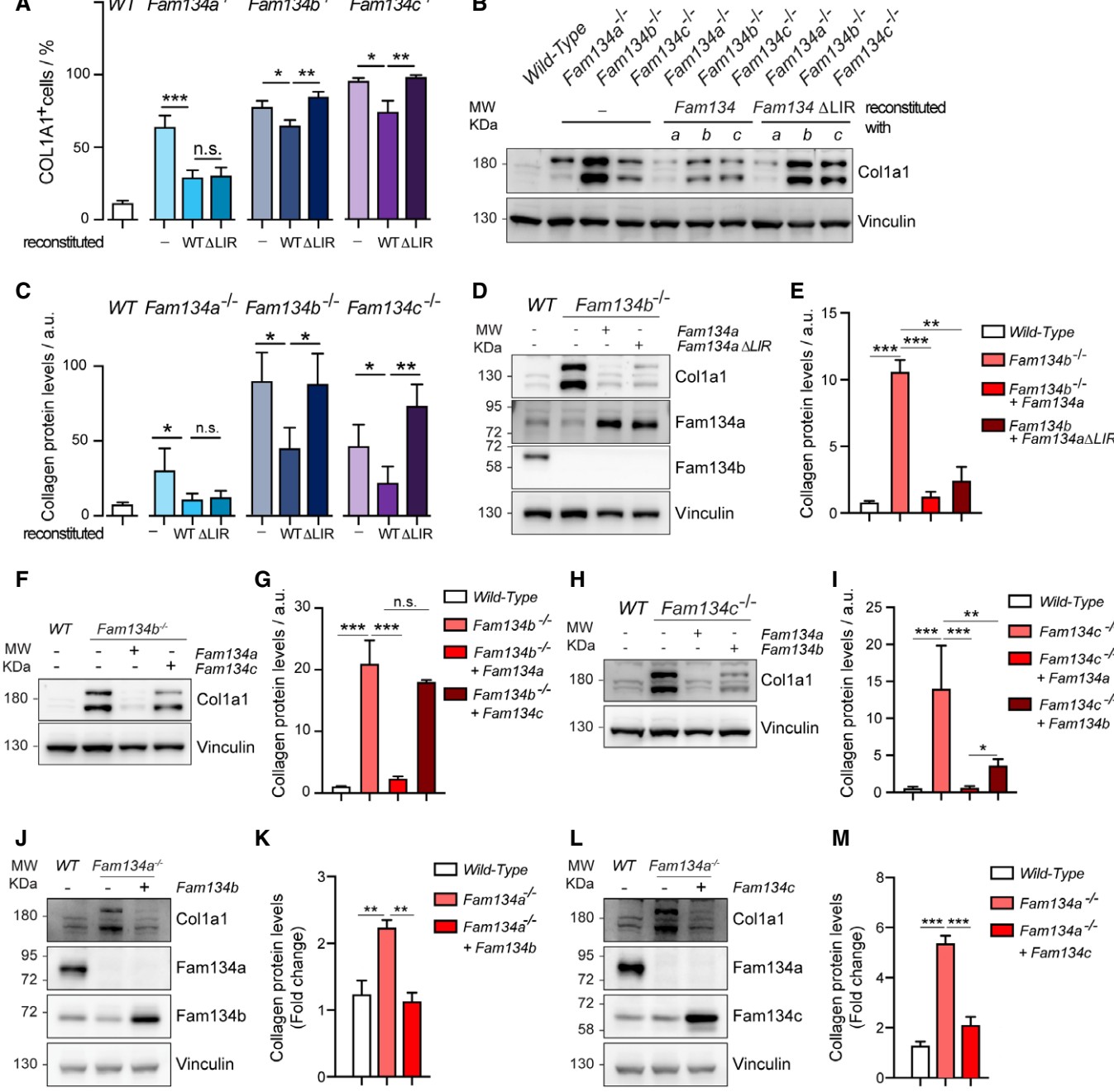

**Figure 7.**

**Figure 7. Absence of Fam134 impacts Collagen I homeostasis.**

A Percentage of cells that are positive for pro-Collagen I in wild-type (WT), *Fam134* knockout MEFs, and *Fam134* knockout MEFs reconstituted with the respective wild-type or ΔLIR mutant Fam134 protein. n = 150 cells each condition.

B, C Representative Western blot (B) of total cell lysate and the corresponding densitometric analysis (C) of Collagen I in wild-type MEFs and *Fam134* knockout MEFs reconstituted for the respective wild-type (WT) or ΔLIR mutant Fam134 protein. Vinculin has been used as reference for ratio calculation. The statistical significance was estimated by unpaired Student's *t* test. All data are represented as mean ± s.e.m. of n = 3 biological experiments, and the statistical significance is defined as *P < 0.05, **P < 0.01, ***P < 0.001.

D, E Representative Western blot (D) and the relative bar plot (E) showing collagen expression in *Fam134b* knockout MEFs overexpressing wild type and the ΔLIR Fam134a protein.

F, G Representative Western blot (F) and the relative bar plot (G) showing collagen expression in *Fam134b* knockout MEFs overexpressing wild-type Fam134a and Fam134c.

H, I Representative Western blot (H) and the relative bar plot (I) showing collagen expression in *Fam134c* knockout MEFs overexpressing wild-type Fam134a and Fam134b.

J, K Representative Western blot (J) and the relative bar plot (K) showing collagen expression in *Fam134a* knockout MEFs overexpressing wild-type Fam134b.

L, M Representative Western blot (L) and the relative bar plot (M) showing collagen expression in *Fam134a* knockout MEFs overexpressing wild-type Fam134c.

Data information: vinculin has been used as reference for ratio calculation. The statistical significance was estimated by ordinary one-way ANOVA. All data are represented as mean ± s.e.m. of n = 3 biological experiments, and the statistical significance is defined as *P < 0.05, **P < 0.01, ***P < 0.001.

Source data are available online for this figure.

that the primary cellular function of FAM134A may be LIR-independent. Structurally, the weak interaction between the FAM134A LIR motif and LC3B could be explained by the absence of a short helical domain following the LIR motif. In yeast, the binding between Atg8 and Atg40, a FAM134B homolog, is mediated by the presence of a helix adjacent to the AIM domain (Mochida *et al*, 2020). Further studies are required to decipher differences in protein complex compositions required for these paralog-specific pathways. Such studies could shed light on the presence of a potential Fam134a co-receptor with an active LIR motif. Alternatively, Fam134a-dependent clearance could represent an alternative delivery pathway for ER components (including Collagen I) to lysosomes (Kamimoto *et al*, 2006; Omari *et al*, 2018; Loi *et al*, 2019).

While there is a growing body of literature on the physiological and pathological roles of FAM134B (Mo *et al*, 2020), FAM134A and FAM134C are still poorly studied. FAM134A has been associated with mitotic progression (Toyoda *et al*, 2017) and is a target of hsa-miRNA940 in the regulation of osteogenic differentiation (Hashimoto *et al*, 2018). FAM134C was reported to regulate neurite outgrowth in human neuroblastoma cells and in primary rat hippocampal neurons (Wang *et al*, 2013). Of note, mutations in the human *FAM134B* gene are responsible for cell death of primary dorsal root ganglion neurons (Kurth *et al*, 2009). Therefore, these two FAM134s are both expressed and seem to have a biological function in cells of the nervous system. Whether and when FAM134 proteins cooperate or act independently to fulfill their physiological functions in different tissues and cellular contexts remains to be determined. Overall, our findings point to the dynamic nature of ER-phagy and the need to better understand how it is regulated. Considering their multifunctional capacity, it is likely that not only the simple activity of Fam134 family members is tightly regulated, but also their abundance, PTMs, and their distribution within the ER network. Likely, the regulation occurs on both the transcriptional and the post-translational level, leading to the oligomerization of ER-phagy receptors or the formation of dynamic complexes with co-receptors. There are probably many more, yet undefined roles of individual FAM134 family members in specific cell types and tissues, leaving us with new challenges to address in future studies.

# Materials and Methods

### Cloning procedures and DNA mutagenesis

cDNAs were cloned into pDONR223 vector using the BP Clonase Reaction Kit (Invitrogen) and further recombined, though the LR Clonase Reaction Kit (Invitrogen), into GATEWAY destination vectors: iTAP MSCV-N-FLAG-HA IRES-PURO, pBABE destination vector (Addgene #51070) and pBABE (Addgene #51070) where we enzymatically changed the resistance from puromycin to hygromycin. cDNA mutations were generated via PCR site-directed mutagenesis according to standard protocols. All cDNAs used in the manuscript are reported in Table EV4.

### Immunoblotting of total cell lysates

Cells were harvested removing culture medium and washed with PBS. Cells were lysed in ice-cold Radio Immuno-Precipitation Assay lysis buffer (150 mM NaCl, 50 mM Tris–HCl pH 7.5, 1% Nonidet P-40 (NP-40), 1 mM EGTA, 5 mM $MgCl_2$, 0.1% SDS) supplemented with protease and phosphatase inhibitors (Roche). Insoluble cell components were separated at 15,500 *g* for 30 min at 4°C and total protein concentration was assessed using Bradford reagent. Protein denaturation was performed in Laemmli buffer at 95°C for 10 min.

Protein lysates were resolved in SDS–PAGE gels and transferred to nitrocellulose (GVS North America, 0.45 μm) or PVDF (Millipore, 0.2 μm) membrane. Membranes were saturated in TBS 0.1% Tween containing 5% low fat milk or BSA and incubated over night at 4°C with the specific primary antibody reported in Table EV5.

### Co-immunoprecipitation

Stable cell lines for FAM134A,B,C were induced with doxycycline (1 μg/ml) for 24 h. Confluent 15-cm-diameter petri dishes of U2OS cells expressing the FAM134s were harvested in 50 mM Tris–HCl (pH 8.0), 120 mM NaCl, 1% (v/v) NP-40, complete protease inhibitor cocktail (Roche). Lysates were cleared by centrifugation at 10,000 *g* for 10 min and incubated over night at 4°C with monoclonal anti-HA-agarose antibody (Thermo Scientific; cat#26181) or

anti CANX antibody (AbCam) that was first conjugated to Protein G Sepharose beads (Thermo Fisher; cat#15918014). Beads were then washed three times in wash buffer [0.1% (v/v) NP-40], resuspended in Laemmli buffer and boiled. Supernatants were loaded on SDS–PAGE and Western blot performed using the indicated antibodies.

### GST pull-down

LC3s, GABARAPs, and Ub proteins were cloned into pGEX-4T-1 vector (GE Healthcare), as GST fusion proteins, and expressed in Escherichia coli BL21 (DE3) cells grown in LB medium (Kirkin *et al*, 2009). Expression was induced by addition of 0.5 mM IPTG, and cells were incubated at 37°C for 5 h. Harvested cells were lysed using sonication in lysis buffer (20 mM Tris–HCl pH 7.5, 10 mM EDTA, 5 mM EGTA, 150 mM NaCl), and GST-fused proteins were immuno-precipitated using Glutathione Sepharose 4B beads (GE Healthcare). Fusion protein-bound beads were used directly in GST pull-down assays. HEK293T cells were transfected with the indicated constructs using GeneJuice (Merck). After 24 h, cells were lysed in lysis buffer [50 mM HEPES, pH 7.5, 150 mM NaCl, 1 mM EDTA, 1 mM EGTA, 1% Triton X-100, and 10% glycerol supplemented with protease inhibitors (Complete, Roche)]. MEFs cells were used for endogenous pull-down of Fam134a,b,c. Lysates were cleared by centrifugation at 12,000 *g* for 10 min, and incubated with GST fusion protein-loaded beads over-night at 4°C. Beads were then washed three times in lysis buffer, resuspended in Laemmli buffer and warmed at 95°C for 5 min. Supernatants were loaded on SDS–PAGE.

### Cell culture

HEK293T were purchased from ATCC. Their identities were authenticated by STR analysis. U2OS TRex cells were provided by Prof. Stephen Blacklow (Brigham and Women's Hospital and Harvard Medical School) (Gordon *et al*, 2009), $Fam134a^{-/-}$, $Fam134b^{-/-}$, and $Fam134c^{-/-}$ MEFs were provided by Prof. Christian Hübner (Jena University). All cell lines were regularly tested negative for the presence of mycoplasma using LookOut Mycoplasma PCR Detection Kit (Sigma). Cells were maintained at 37°C with 5% $CO_2$ in DMEM medium (Thermo Scientific) supplemented with 10% fetal bovine serum (Thermo Scientific) and 100 U/ml penicillin and streptomycin (Thermo Scientific). Starvation was conducted by incubating cells in EBSS medium (Thermo Scientific). U2OS stable and inducible cells were treated with 1 μg/ml of doxycycline (Sigma) to induce FAM134 expression and with 200 ng/ml bafilomycin A1 (Aurogene) to block autophagy. ER-phagy was induced with EBSS or 250 nM Torin1 (Tocris) treatments. For each treatment, cells were plated the day before in order to perform the experiments when cells had a final confluency of 70-80%. For transient expression, DNA plasmids were transfected with GeneJuice (Merck) or Turbofect (Thermo Scientific) according to the instructions of the manufacturers. Stable cell lines were produced using retroviral or lentiviral virus infection. U2OS TRex stable and inducible cell lines were generated cloning the cDNAs into MSCV iTAP N-FLAG-HA retroviral vector. *Fam134a,b,c* knockout MEFs (obtained from *Fam134a,b,c* single knockout mice generated in Prof. Hübner's Laboratory) have been reconstituted with cDNAs cloned in retroviral pBABE plasmid. Retrovirus were generated in HEK293T cells

co-transfecting the retroviral vector, containing the selected cDNA, with the packaging plasmid (viral Gag-Pol) and the envelope plasmid (VSV-G). Lentivirus were also generated co-transfecting in HEK293T cells with the ssRFP-GFP-KDEL vector together with pPAX2 packaging plasmid and pMD2.G envelope plasmid.

### Antibodies used for Western blot, immuno-precipitation, and immuno-fluorescence staining

A complete list of primary antibodies used for the experiments is reported in Table EV5. Secondary antibody, HRP-conjugated, was purchased from Millipore (rabbit IgG #12348, mouse IgG #12349, rat IgG PI-9400-1). For immuno-fluorescence staining, the following antibodies were used: anti-rabbit Alexa Fluor® 555 (Thermo Scientific; A31572), anti-rabbit Alexa Fluor® 647 (Thermo Scientific; A21244), anti-rat Alexa Fluor® 647 (Thermo Scientific; A21472), anti-mouse Alexa Fluor® 647 (Thermo Scientific; A21236), anti-rat Alexa Fluor® 488 (Thermo Scientific; A21208), and anti-rabbit Alexa Fluor® 405 (AbCam; ab175660).

### Fluorescence microscopy

Cells were plated on glass cover slips, for at least 24 h, and then fixed with 4% paraformaldehyde for 10 min. Cells were permeabilized with a 0.1% Saponin solution in PBS at room temperature for 5 min and then blocked in PBS containing 10% fetal bovine serum (FBS) for an additional hour at room temperature. Cells were incubated over night at 4°C with primary antibody diluted in PBS containing 1% FBS and 0.1% Saponin. Washes were performed in 0.1% Saponin in PBS. Cells were incubated with secondary antibodies for 30 min at room temperature and then washed two times with PBS 5% FBS, 0.1% Saponin and once with PBS. The coverslips were mounted on with an aqueous mounting medium (Mowiol) and placed on a glass holder. Images were acquired with a Zeiss LSM800 and a Zeiss LSM880 laser-scanning microscope (Zeiss). Images shown are representative of experiments carried out at least three times.

### Electron microscopy

For pre-embedding immuno-EM, U2OS cells stably-expressing HA-FAM134s were fixed with a mixture of 4% paraformaldehyde (PFA) and 0.05% glutaraldehyde (GA) prepared in 0.2 M HEPES for 10 min, then fixed with 4% PFA alone for 30 min, followed by an incubation with a blocking–permeabilizing solution (0.5% BSA, 0.1% Saponin, 50 mM $NH_4Cl$) for 30 min. Primary anti-HA antibody (Covance) and 1.4-nm gold-conjugated Fab' fragments of anti-mouse immunoglobulin G (Nanoprobes, Yaphank, NY, cat N° 2002) were diluted in blocking–permeabilizing solution and added to the cells overnight and for 2 h, respectively. The GoldEnhance EM kit (Nanoprobes, Yaphank, NY, cat N° 2113) was used to enhance the ultrasmall gold particles. The cells were scraped, pelleted, post-fixed in $OsO_4$ and uranyl acetate, dehydrated in ethanol and embedded in Epon. For routine EM, MEFs were fixed with 1% GA prepared in 0.2 M HEPES for 30 min, then scraped, pelleted, post-fixed and embedded as indicated earlier. From each sample, thin 60-nm sections were cut using a Leica EM UC7 (Leica Microsystems, Wetzlar, Germany). EM images were acquired from thin sections

under an EM (Tecnai G2 Spirit BioTwin; FEI; https://www.fei.com/) equipped with a VELETTA CCD digital camera (Soft Imaging Systems GmbH, Munster, Germany).

## Mass spectrometry

All the experiments were performed in label-free settings. Proteins were precipitated in acetone and then reduced and alkylated in a solution of 6 M Guanidine-HCl, 5 mM TCEP, and 20 mM chloroacetamide. Peptides were obtained digesting proteins with LysC (WAKO) for 3 h at 37°C and with the endopeptidase sequencing-grade Trypsin (Promega) overnight at 37°C. Collected peptide mixtures were concentrated and desalted using the Stop and Go Extraction (STAGE) technique (Rappsilber et al, 2003). Instruments for LC MS/MS analysis consisted of a NanoLC 1200 coupled via a nano-electrospray ionization source to the quadrupole-based Q Exactive HF benchtop mass spectrometer (Michalski et al, 2011). Peptide separation was carried out according to their hydrophobicity on a home-made chromatographic column, 75 μm ID, 8 Um tip, 250 mm bed packed with Reprosil-PUR, C18-AQ, 1.9 μm particle size, 120 Angstrom pore size (New Objective, Inc., cat. PF7508-250H363), using a binary buffer system consisting of solution A: 0.1% formic acid and B: 80% acetonitrile, 0.1% formic acid. Runs of 120 min after loading were used for proteome samples, with a constant flow rate of 300 nl/min. After sample loading, run start at 5% buffer B for 5 min, followed by a series of linear gradients, from 5% to 30% B in 90 min, then a 10 min step to reach 50% and a 5 min step to reach 95%. This last step was maintained for 10 min. Q Exactive HF settings: MS spectra were acquired using 3E6 as an AGC target, a maximal injection time of 20 ms, and a 120,000 resolution at 200 m/z. The mass spectrometer operated in a data-dependent Top20 mode with sub sequent acquisition of higher-energy collisional dissociation (HCD) fragmentation MS/MS spectra of the top 20 most intense peaks. Resolution, for MS/MS spectra, was set to 15,000 at 200 m/z, AGC target to 1E5, max injection time to 20 ms and the isolation window to 1.6Th. The intensity threshold was set at 2.0 E4 and Dynamic exclusion at 30 s.

## ER-phagy assay

U2OS TRex FLAG-HA-FAM134s stable and inducible cell lines were infected with lentivirus carrying the pCW57-CMV-ssRFP-GFP-KDEL (Addgene #128257). We previously deleted the tetracycline response element, via PCR, from ssRFP-GFP-KDEL in order to have constitutive expression. Wild-type and *Fam134* knockout MEFs were infected with pCW57-CMV-ssRFP-GFP-KDEL (Addgene #128257) in order to generate ER-phagy reporter inducible cell lines. Knockout MEFs were further reconstituted with the wild-type and LIR mutant forms of the corresponding FAM134 cloned in the pBABE plasmid (Addgene #51070) where we switched puromycin resistance to hygromycin. Fluorescence of GFP and RFP as well as cell confluence (phase) was monitored over time via the IncuCyte S3 (Sartorius, Germany) in 384-well format. 1,500 cells (U2OS) and 1,300 cells (MEFs) per well were seeded in 50 μl DMEM media, supplemented with 10% Fetal Bovine Serum (FBS) 1% Penicillin/Streptomycin 2 μg/ml Puromycin 15 μg/ml Blasticidin 1 μg/ml doxycycline or 50 μl DMEM media plus 10% FBS 1% Penicillin/Streptomycin 2 μg/ml puromycin 100 μg/ml hygromycin 1 μg/ml doxycycline, respectively, and incubated for 24 h before treatment. For indicated treatments, 50 μl of media containing either 0.2% DMSO, 500 ng/ml Torin1 or the indicated concentrations of Thapsigargin were added. Between the actual treatment and the first scan point (0 h), appx. 5–10 min passed for technical reasons. Screens in all three channels were taken at indicated time points following the treatment. ER-phagy flux was monitored through changes in the ratio of the total fluorescence intensity of RFP/GFP. Each point represents the averaged ratio of data obtained from three individual wells (technical replicates), and experiments have been performed in three biological replicates with comparable results.

## Structural modeling for FAM134 family members

Previously, we built a molecular model of FAM134B-RHD by integrating fragment-based modeling with extensive molecular dynamic (MD) simulations (Bhaskara et al, 2019). We tested and validated this model by MD simulations, in-cell functional assays, and in vitro membrane remodeling assays (Bhaskara et al, 2019). Therefore, we used the FAM134B-RHD structure as a template for modeling the RHDs of FAM134A and FAM134C. We used an alignment method to score the hydrophobicity patterns (AlignME), and to generate two pair-wise alignments and map the topologically equivalent positions (TM-segments and AH-helices) onto the template structure (Stamm et al, 2014; Tsirigos et al, 2015). We then used Modeller to build the 3D structures of FAM134A-RHD and FAM134C-RHD (Sali & Blundell, 1993).

## Molecular dynamics simulations and analysis

We performed coarse-grained MD simulations using the MARTINI model (version 2.2) (Marrink et al, 2007; Monticelli et al, 2008). Initial CG structures of the FAM134 family members were built by using *martinize.py* (Wassenaar et al, 2015). DSSP assignments were used to generate backbone restraints to preserve the local secondary structure (Jones, 1999). CG models were embedded into POPC (16:0–18:1 PC) bilayers spanning the periodic simulation box in the *xy* plane. Initial configurations for each system were assembled and then solvated with CG-water containing 150 mM NaCl using *insane.py* script (Wassenaar et al, 2015). Each system was first energy minimized and equilibrated using the Berendsen thermostat and barostat along with position restraints on protein backbone beads followed by production runs with a 20-fs time step. System temperature and pressure during the production phase were maintained at 310 K and 1 atm with the velocity rescaling thermostat and the semi-isotropic Parrinello-Rahman barostat, respectively. All simulations were performed using gromacs (version 2019.3; Pronk et al, 2013). The RHD shape was characterized by computing the radius of gyration $R_g$ using all protein CG-beads along the trajectory to draw probability densities. Long-lived, highly populated RHD conformations were obtained after clustering evenly sampled conformations ($n = 10,000$) from each trajectory. Clusters were obtained using backbone RMSD (cutoff = 0.8 nm) by employing the *gromos* method, as implemented in the *gmx_cluster* tool. Internal protein dynamics along the RHD was quantified by measuring the root mean square fluctuations (RMSF) of atomic positions from structures sampled along the trajectory.

## Bicelle-to-vesicle transitions and kinetics

Discontinuous bicelle systems containing saturated DMPC (14:0 PC) and DHPC (7:0 PC) lipids were assembled using a protocol established previously. The equilibrated RHD conformations obtained from simulations in a POPC bilayer (after 5 μs) were embedded in the bicelle and solvated. Twenty replicates for each system were simulated with different initial velocities to obtain statistics on the transition times to vesicles. Shape transformations from flat bicelles ($H = 0$ nm$^{-1}$) to curved vesicles ($H = 0.15$ nm$^{-1}$) were monitored by measuring the signed membrane curvature ($H(t)$). Lipid coordinates were fitted to spherical surfaces using least-squares optimization to compute membrane curvature along simulations using MemCurv (https://github.com/bio-phys/MemCurv). Curvature away from and toward the upper/cytoplasmic leaflet are reported as positive and negative values, respectively. The statistics of waiting times ($t$) for the formation of vesicles (bilayer curvature, |H| > 0.15 nm$^{-1}$) for the three systems were determined from individual replicates. The kinetics of the bilayer-to-vesicle transition was modeled using a Poisson process with a lag time ($t = t' + \tau$). The time $t' = 1/k'$ describes the Poisson process with rate $k'$. The constant lag time $\tau$ captures the time required for vesicle closure from the curved bilayer-disk. The distributions of waiting times are thus $p(t) = k'\ e^{-k'(t-\tau)}$ for $t > \tau$. We determined the rate of vesicle formation ($k$ vesicle $= 1/(t' + \tau)$) for different systems, from fitting the cumulative distribution function for the probability density, $p(t)$, to the observed waiting time distributions estimated from replicates. We used the previously computed maximum likelihood estimates of the vesiculation rate for empty bicelles ($k_{\text{Empty}} = 1.09 \times 10^{-5}$) to compute the acceleration factors for each system (acc $= k_{\text{sys}}/k_{\text{Empty}}$) (Bhaskara *et al*, 2019).

## Data analysis

All experiments were performed in at least three independent biological replicates. Data are presented as mean ± s.e.m. Statistical analysis between two experimental groups was performed using parametric two-tailed Student's *t* test. For immuno-staining quantifications, 150–200 cells were analyzed for each experimental condition in each replicate. For autophagy flux experiments, data obtained from one scan containing > 1,000 cells of three different wells with indicated treatment were analyzed. For Western blot quantifications, densitometric analysis were performed considering three independent blot imagines.

For mass spectrometry, all acquired raw files were processed using MaxQuant (1.6.2.10) and the implemented Andromeda search engine. For protein assignment, spectra were correlated with the UniProt Homo Sapiens and Mus Musculus database (v. 2019) including a list of common contaminants. Searches were performed with tryptic specifications and default settings for mass tolerances for MS and MS/MS spectra. Carbamidomethyl at cysteine residues was set as a fixed modification, while oxidations at methionine and acetylation at the N-terminus were defined as variable modifications. The minimal peptide length was set to seven amino acids, and the false discovery rate for proteins and peptide-spectrum matches to 1%. The match-between-run feature with a time window of 0.7 min was used. For further analysis, the Perseus software (1.6.2.3) was used and first filtered for contaminants and reverse entries as well as proteins that were only identified by a modified peptide. The LFQ ratios were logarithmized, grouped, and filtered for minutes valid number (minutes 3 in at least one group). Missing values have been replaced by random numbers that are drawn from a normal distribution. Significantly regulated proteins between conditions were determined by ANOVA (FDR < 0.01). Hierarchical clustering was performed on significant proteins, using Fisher's exact test to calculate enrichments in categorical terms, setting Benjamin-Hochberg FDR < 0.05 as threshold. The protein–protein interaction network was built in the cytoscape environment (Shannon *et al*, 2003). Proteins belonging to the selected cluster were loaded into the STRING plugin, and the network was subsequently generated (Jensen *et al*, 2009). Data visualization was done in the statistical environment R. MS analyses of three independent samples for each experiment were performed. Pearson's correlation coefficients up to 0.9 indicated the high reproducibility between biological repetitions.

# Data availability

The mass spectrometry proteomics data have been deposited to the ProteomeXchange Consortium via the PRIDE partner repository with the dataset identifier PXD021690 (http://www.ebi.ac.uk/pride/archive/projects/PXD021690) (MEFs proteomes) and PXD025626 (http://www.ebi.ac.uk/pride/archive/projects/PXD025626) (U2OS proteomes).

**Expanded View** for this article is available online.

## Acknowledgements

We thank Ivan Dikic (IBC2, Frankfurt am Main, Germany) for suggestions and critical comments on the manuscript. We acknowledge Anna Vainshtein (Craft Science Inc.) for critical reading and Chiara De Leonibus for technical support. We thank the microscopy and mass spectrometry facilities at TIGEM Institute. This work was supported by grants to PG: Telethon Foundation (TMPGCBX16TT), Roche Foundation (Roche per la Ricerca 2019), AFM-Telethon (Trampoline Grant 2020); and to AS: Deutsche Forschungsgemeinschaft (DFG, German Research Foundation)—Project-ID 259130777—SFB 1177, the Else Kroener Fresenius Stiftung (2016_A196), and the EU/EFPIA/OICR/McGill/KTH/ Diamond Innovative Medicines Initiative 2 Joint Undertaking (EUbOPEN grant n° 875510). AR is supported by Fondazione Umberto Veronesi. CAH was funded by the DFG (HU 800/6-2 and RTG 1715). RMB and GH thank the Max Planck Society for support. Open Access funding enabled and organized by Projekt DEAL.

## Author contributions

AR performed most of the experiments and generated the majority of the cell lines. VB performed GST pull-downs and generate cell lines. RB performed some of the IPs and with support of MT performed the ER-phagy assays. RMB developed the computational methodology, performed structural modeling, molecular dynamics simulation, and analysis of simulation data with support of GH. RMB also contributed to the writing of the modeling and simulation sections. IP generated the plasmids. EP performed EM analysis. GDL generated cell lines. CC analyzed mass spectrometry data. ME performed Western blot. AH isolated MEFs from mice and AKH generated *Fam134* knockout mice. CS and CAH provided tools and critical comments. PG and AS designed the study, supervised the experiments and jointly wrote the manuscript. All the authors read and commented on the final manuscript.

## Conflict of interest

The authors declare that they have no conflict of interest.

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
