## [Review Process File · EMBO Reports]

Role of FAM134 paralogues in endoplasmic reticulum remodeling, ER-phagy and Collagen quality control

Alessio Reggio, Viviana Buonomo, Rayene Berkane, Ramachandra Bhaskara, Mariana Tellechea, Ivana Peluso, Elena Polishchuck, Giorgia Di Lorenzo, Carmine Cirillo, Marianna Esposito, Adeela Hussain, Antje Hübner, Christian Hübner, Carmine Settembre, Gerhard Hummer, Paolo Grumati, and Alexandra Stolz

DOI: 10.15252/embr.202052289

Corresponding author(s): Alexandra Stolz (stolz@em.uni-frankfurt.de) , Paolo Grumati (p.grumati@tigem.it)

Review Timeline:

Submission Date:	16th Dec 20
Editorial Decision:	22nd Dec 20
Revision Received:	26th Apr 21
Editorial Decision:	18th Jun 21
Revision Received:	27th Jun 21
Editorial Decision:	6th Jul 21
Revision Received:	7th Jul 21
Accepted:	8th Jul 21

Transaction Report: This manuscript was transferred to EMBO reports following peer review at The EMBO Journal.

Dear Alexandra,

Thank you for the transfer of your manuscript and the associated referee reports from our sister journal The EMBO Journal to EMBO Reports. As discussed, we would like to invite you to revise your manuscript for potential publication in EMBO Reports along the lines you outlined in your point-by-point response.

It will be important to better document and explain the overlapping and distinct functions of FAM134A and -C as well as the LIR-dependent and -independent roles of FAM134A in the regulation of ER morphology and collagen handling. Cell-type specific effects (MEFs, U2OS) should be revisited and clarified, at least in the text and discussion.

It will not be necessary to analyse triple knockout cells, to provide experimental support for the MD simulations or to investigate potential post-translational modifications that activate the FAM134 receptors.

Please address all referee concerns in a complete point-by-point response. Acceptance of the manuscript will depend on a positive outcome of a second round of review. It is EMBO reports policy to allow a single round of revision only and acceptance or rejection of the manuscript will therefore depend on the completeness of your responses included in the next, final version of the manuscript.

We invite you to submit your manuscript within three months of a request for revision. This would be March 22nd in your case. However, we are aware of the fact that many laboratories are not fully functional due to COVID-19 related shutdowns and we have therefore extended the revision time for all research manuscripts under our scooping protection to allow for the extra time required to address essential experimental issues. Please contact us to discuss the time needed and the revisions further.

- 1) A data availability section is missing.
- 2) Your manuscript contains error bars based on $n=2$. Please use scatter blots showing the individual datapoints in these cases. The use of statistical tests needs to be justified.

2) individual production quality figure files as .eps, .tif, .jpg (one file per figure).

Please download our Figure Preparation Guidelines (figure preparation pdf) from our Author Guidelines pages

<https://www.embopress.org/page/journal/14693178/authorguide> for more info on how to prepare your figures.

4) a complete author checklist, which you can download from our author guidelines (). Please insert information in the checklist that is also reflected in the manuscript. The completed author checklist will also be part of the RPF.

5) Please note that all corresponding authors are required to supply an ORCID ID for their name upon submission of a revised manuscript (). Please find instructions on how to link your ORCID ID to your account in our manuscript tracking system in our Author guidelines ()

6) We replaced Supplementary Information with Expanded View (EV) Figures and Tables that are collapsible/expandable online. A maximum of 5 EV Figures can be typeset. EV Figures should be cited as 'Figure EV1, Figure EV2' etc... in the text and their respective legends should be included in the main text after the legends of regular figures.

7) Before submitting your revision, primary datasets (and computer code, where appropriate) produced in this study need to be deposited in an appropriate public database (see < <https://www.embopress.org/page/journal/14693178/authorguide#dataavailability>>).

Specifically, we would kindly ask you to provide public access to the proteomics datasets.

The accession numbers and database should be listed in a formal "Data Availability " section (placed after Materials & Method) that follows the model below (see also < <https://www.embopress.org/page/journal/14693178/authorguide#dataavailability>>). Please note that the Data Availability Section is restricted to new primary data that are part of this study.

Data availability

- RNA-Seq data: Gene Expression Omnibus GSE46843

(<https://www.ncbi.nlm.nih.gov/geo/query/acc.cgi?acc=GSE46843>)

- [data type]: [name of the resource] [accession number/identifier/doi] ([URL or

identifiers.org/DATABASE:ACCESSION])

8) We would also encourage you to include the source data for figure panels that show essential data. Numerical data should be provided as individual .xls or .csv files (including a tab describing the data). For blots or microscopy, uncropped images should be submitted (using a zip archive if multiple images need to be supplied for one panel). Additional information on source data and instruction on how to label the files are available .

10) Regarding data quantification

- the name of the statistical test used to generate error bars and P values,
- the number (n) of independent experiments (please specify technical or biological replicates) underlying each data point,
- the nature of the bars and error bars (s.d., s.e.m.)

11) As part of the EMBO publication's Transparent Editorial Process, EMBO reports publishes online a Review Process File to accompany accepted manuscripts. This File will be published in conjunction with your paper and will include the referee reports, your point-by-point response and all pertinent correspondence relating to the manuscript.

I look forward to seeing a revised version of your manuscript when it is ready. Please let me know if you have questions or comments regarding the revision.

Kind regards,

Martina

Martina Rembold, PhD

Senior Editor
EMBO reports

Referee #1:

Comments EMBO J

Critical functions of FAM134 family members in endoplasmic reticulum remodeling, ER-phagy and collagen quality control.

In this manuscript, Reggio and co-workers have identified two new ERphagy receptors, FAM134A and FAM134C, both being similar to the already described FAM134B in that they contain a LIR motif that is required for their ERphagy function. However, in contrast to FAM134B, which induces ERphagy upon overexpression, FAM134A and -C mediated ERphagy require an activation signal. All FAM134s were found to interact with Calnexin and affect pro-collagen I levels, and although they have somewhat redundant functions when it comes to regulation of collagen I turnover, FAM134A seems to regulate this in an LIR-independent manner.

The results presented in this manuscript are generally well done, but unfortunately not of sufficient quality to support the conclusions drawn. The identification of FAM134A and FAM134C as ERphagy receptors is novel, but more insight into the mechanisms involved in regulation of their activity and how this differ from FAM134B under various (patho)physiological conditions are required to warrant publication in EMBO J. The authors should consider the specific comments below.

General comments:

1. The main shortcoming with this manuscript is the fact that all ERphagy data is based on overexpression of the different human FAM134s in U2OS cells (no knock-out or rescue), while their effects on collagen I levels are done in KO MEFs cells with rescue of mouse Fam134s. To be able to conclude they should do the ERphagy assays in KO-rescue cells (U2OS and/or MEFs).
2. When assessing effects of the various FAM134s on ERphagy, they generally use the overexpressed proteins themselves as a read-out (e.g. in Figure 2, EV3). It would be important to also quantify ERphagy with other ER markers to make sure they are not simply looking at different rates of turnover of the FAM134 proteins. It is interesting that they did not have any western blot data of ER proteins with knockdown/overexpression of the FAMs, e.g. in EV5C, although it is a Co-IP experiment, it looks like Calnexin is not altered in the cells overexpressing the FAMs. Maybe they can also highlight ER resident proteins in their FAM KO mass specs to support Figure 2.
3. It is nice that they in Figure 3A use the ssRFP-GFP-KDEL ER-phagy reporter, but again only for cells overexpressing the different FAMs. They should do similar experiments with depletion of the different FAM134s +/- induction of ERphagy to show that these proteins are really required for the Torin-induced ERphagy. Assessing effects of overexpression, as done in Figure 3, is not very physiological relevant (could be overexpression artefacts). Rather they should do rescue experiments in a knock-out background.
4. Based on in silico work with the RHD domains of the FAM134s, the authors propose a model where the RHDs of FAM134B and FAM134C are more dynamic than the RHD of FAM134A, resulting in differences in dynamics and curvature induction capacities of the

respective RHDs, thus providing an explanation for why FAM134B and to a lesser degree C) are better at causing ER fragmentation. This is nice, but as there is no experimental evidence directly supporting this model (e.g. mutation in the respective RHDs), this is highly speculative.

5. To be able to conclude that the various FAM134s have a different role in basal vs starvation-induced ERphagy it is important that the expression levels of the various FAM134 proteins are similar. How do the levels of HA-FAM134s when expressed in U2OS cells compare to each other and to endogenous FAM134 levels? In fig. EV5, one can see some differences in the expression levels of the different HA-FAM134s wt and LIR mutants. More worrying is the presence of several extra bands for the HA-FAM134s that can be seen in EV5 (especially upon high exposure E), where some do not correspond to the bands seen in EV5C,D. Why is the MW of FAM134B different in these Ips??

6. Page 5: the heading "FAM134A and FAM134C can drive ER fragmentation" is somewhat misleading, as they do not seem to drive fragmentation on their own. I would rather write ""FAM134A and FAM134C are required for starvation-induced ER fragmentation". Torin and starvation may not be the most specific ER-phagy inducers, maybe they can also try cyclopiazonic acid (CPA).

7. Since overexpression of FAM134A and C did not induce ER-phagy without starvation, a signal (like PTM) is probably required to activate these receptors. Are these proteins modified by phosphorylation or ubiquitination?

8. The authors argue that Fam134c seems to act as a facilitator of Fam134b activity in the degradation of misfolded collagen, while Fam134a mediates a parallel LIR-independent degradation pathway. This is based on largely descriptive data and more data is needed to conclude about the mechanisms involved, especially as the role of FAM134A in shaping ER morphology requires a functional LIR motif. The data are also a bit confusing; in EV6A, it appears that the FAMs are not redundant as depletion of any one of the FAMs led to increase in collagen accumulation. However, in EV6D, Fam134a was able to rescue from collagen accumulation in Fam134b knockout, suggesting that Fam134a and Fam134b are redundant (in contrast to EV6A).

9. In general, the figures should be more or less self-explanatory. This is not always the case here and the authors should pay specific attention to this. E.g. in figure 1C, they should write HA-FAM134 above image so that it is clear that HA in inset refers to overexpressed (not endogenous) FAM134.

10. Figure legends should describe what was done and not the results, as is sometimes done, e.g. in Figure 2B; "Wild-Type FAM134s specifically bind to GST-LC3B and GSTGABARAPL1 via their LIR domain. FAM134s do not interact with GST-Ubiquitin and GST-Tetra-Ubiquitin».

Specific comments:

1. Figure 1B demonstrates nicely that both FAM134A and -C contain a functional LIR motif, but why only test binding to LC3B and GABARAPL1? They should test binding to the other ATG8 proteins as well. The set-up and labelling of this figure are confusing. They should indicate in the figure that this is using myc-FAM134s and GST-LC3B/GAB-L1. The empty

should be GST? What is the difference between the input samples to the right vs the left part of the figure? The Ponceau staining must be from one of the pulldowns? LC3B and GAB-L1 run at different sizes in the two Poncea blots.

2. Figure 1C: labeling of CANX in upper red image missing. It would be better to show a small region of the merge image as insets in different colors (instead of zoom out of same cell(s)). Why use different cells and constructs for Figure 1B (transient transfection of myc-FAM134s in HEK293 cell) and C (stable inducible U2OS HA-FAM134s)?
3. Figure 2: A-B. label all small images with HA and LC3B. Indicate in merge image where inset is taken from. C. should write Basal+BafA1 and EBSS + BafA1 for quantifications. They need to indicate how many cells were used for quantification and from how many experiments.
4. Figure 3A: should indicate in figure what is red/green (ssRFP-GFPKDEL) and use better quality images. Scale bar is missing.
5. Figure 3B: the legend should describe how the quantifications were done, how many cells, statistics etc. Why is the GFP/RFP ratio at 0h not always 1.0 (varies from 0.8 to 1.2)? What is the girl on the a-axis indicating? It is not so clear how the data in C differ from B. Are there significant differences between FAM134A, -B and C-??
6. In Figure 7A they present "quantification of cells with misfolded pro-collagen I accumulation in wild-type, Fam134a,b,c single knockout MEFs and FAM134a,b,c single knockout cells reconstituted with the wild-type or LIR mutant form of the respective Fam134.", but it doesn't say how this is done, by quantification of images as shown in EV5B? Quantification of collagen intensity or number of cells showing any collagen staining? How many cells, how many experiments?
7. There is no Figure 7J although described in legend.
8. The text could benefit from thorough editing for grammar.
9. The introduction is a bit too extensive.

Referee #2:

The authors compared the activity of FAM134A, B, C in ER-phagy using three different readouts, ER remodeling, collagen quality control and ER-membrane degradation. The major finding of this work is that FAM134A appears to function different from FAM134B and FAM134C, and surprisingly, independent of its interaction with LC3s.

This is potentially an interesting paper suitable in scope and importance for EMBO J. However, currently some of the data are not convincing enough to support their conclusions. Substantial revisions are necessary for publication in EMBO.

Major points:

1. Because FAM134A, FAM134B and FAM134C may interact (through their RTN domains) or affect each other in ER-phagy regulation, the best experimental design to compare their activities would be performing the rescue assays by expressing each individual WT proteins or LIR-deletion mutants in the triple-knockout cells, or at least, in the triple-knockdown cells.
2. It is intriguing that FAM134A mediates pro-collagen degradation without LC3 interaction. Does this mean that FAM134a promotes procollagen degradation independent of the canonical ER-phagy? what is the mechanism? In Figure 5D, both FAM134A Δ LIR and FAM134C Δ LIR failed to rescue ER-phagy activities. It would be contradictory to the major conclusion of this work, if FAM134a Δ LIR drives the degradation of procollagen via macroautophagy pathway. Macroautophagy-independent degradation of ER components has been proposed. The authors may test the possibilities from other perspectives.

Minor points:

1. The statement that "Here, we introduce FAM134A/RETREG2 and FAM134C/RETREG3 as new ER-phagy receptors." in the abstract was inappropriate, because FAM134a and FAM134b were implied as the ER-phagy receptors in previous study(Khaminets et al., 2015).
2. The FAM134C marker standard is not consistent in Figure 5E.
3. The resolution of immunofluorescence image should be improved. The co-localization of FAM134 and CANX in Fig1 was weak.
4. In Fig2C, statistic results do not match the representative images. It is obviously that HA dots were fewer in FAM134C (WT) than FAM134A and FAM134B after treatment with EBSS.
5. "Although the LIR dependent activity of FAM134A and FAM134C does affect ER membranes fragmentation, it did not significantly influence ER tubular structure (ER branching), neither under basal nor starvation conditions". After treatment with EBSS, FAM134A and FAM134C significantly fragmented ER, ER theoretically would be degraded by ER-phagy. Why ER branching stay unchanged? Fig2A and B should be stained with ER marker.
6. Fig.EV3C, quality of EM image should be improved. Gold particle labeling for FAM134A was not specific.
7. Compounds that induce ER stress can be used to induce ER-phagy and detect whether FAM134s protein is degraded, including ER marker.
8. In Fig5C, the effect of Fam134c *-/-* reconstituted with WT protein is better than Fam134a, but was contradictory to the statistic results. And why not use Fam134b *-/-* as control?
9. "Moreover, as previously reported for Fam134b (Forrester et al., 2019), Fam134a and Fam134c co-immuno-precipitate with Canx (Fig.EV5CE). This suggest that Canx may be working as co-receptor between mis-folded procollagen and the Fam134 family of ER-phagy receptors". whether the degradation of pro-collagen is affected in Canx knockdown cell?
10. In Fig.EV6A, Fam134b KO and Fam134b KO reconstituted with deltaLIR, Fam134b KO reconstituted with WT and Fam134c KO reconstituted with WT, the statistical data of these two groups was contradictory to western blot bands in Fig7B.
11. In Fig7I, it was concluded that "Fam134a over-expression had a stronger effect compared to that of Fam134b". In fact, Fam134a overexpression was much higher than Fam134b in Fig7H.

Khaminets, A., Heinrich, T., Mari, M., Grumati, P., Huebner, A.K., Akutsu, M., Liebmann, L., Stolz, A., Nietzsche, S., Koch, N., et al. (2015). Regulation of endoplasmic reticulum turnover by selective autophagy. *Nature* 522, 354-358.

Referee #3:

Reggio and co-authors delineate FAM134A/RETREG2 and FAM134C/RETREG3 as new ER-phagy receptors in this manuscript. The authors compare FAM134A and FAM134C with their family member FAM134B, a well-known ER-receptor, and show that FAM134A and FAM134C require an activation signal to induce ER fragmentation, while FAM134B is sufficient to induce massive ER fragmentation in baseline conditions. They show that the three FAM134 family members display discrete conformations and ability to induce vesicle formation. All Fam134 family members are required to maintain ER homeostasis and are involved in the regulation of pro-collagen I levels. However, the authors reveal a specific function for FAM134a which exerts its function through LIR independent (misfolded

collagen I control) and LIR dependent (ER morphology) mechanisms. Overall the MS is clear and the conclusions that the authors draw are well supported by the data. Also the characterization of FAM134A and FAM134C as new ER-phagy receptors is novel and brings new light into the field. However, there are some controls that should be included and some points that need to be reinforced.

Major points:

The authors claim that mutation of the LIR motif in FAM134A and FAM134C are sufficient to abolish FAM134A/C-LC3B interaction, thus supporting the presence of a functional, canonical LIR motif in FAM134A and FAM134C. However, there is a faint FAM134A band in the blot where LC3B- FAM134A Δ LIR possible interaction is analysed (Figure 1B right panel). This blot should be improved. Moreover, does empty refer to cells transfected with pGEX-4T-1 plasmid? What is the band observed in Panceau image? Authors should annotate this better.

FAM134C images of EBSS condition showed in figure 2A do not match quantification showed in 2C. Quantification shows a higher number of HA dots in FAM134C than in FAM134B overexpression while pictures do not reflect that. Maybe authors could point with an arrow what they mean by HA dots. Does the quantification graph show a representative experiment or a pool of different experiments? Has any statistical test been performed for this quantification?

Why Torin 1 treatment induces specifically ER-phagy? Please explain it briefly.

In their proteomics analysis, can the authors detail whether they used all the proteins for hierarchical clustering and what metric and linkage methods they have used for the same? It is also possible to perform hierarchical clustering on the first 2 components of the PCA, to identify proteins defining each class for those components.

Single knockout MEFs, compensatory increase in protein levels of the remaining two FAM134s were not observed (Figure 5A). However, Fam134c^{-/-} MEFs seem to have lower levels of Fam1234b than WT MEFs. Could authors comment on that? Then authors claim that at least in MEFs, all three Fam134 family members have equally important roles in maintaining proper pro-collagen I quality control. They should downregulate Fam1234 members in another cell line, such as U2OS, and measure pro-collagen I levels by western blot and/or perform Colla1+Lamp1 immunofluorescence to validate the mechanism in a second cell line and rule out the possibility of cell-specific effects.

It seems that knockout MEFs reconstituted with Fam134a rescue completely pro-collagen I accumulation, while reconstitutions with Fam134b and Fam134c show a partial rescue, there is still some pro-collagen I accumulation (Fig.EV5B and Figure 7A). Authors should comment on that.

Over-expression of Fam134a in Fam134b and Fam134c knockout MEFs is sufficient to fully rescue the pro-collagen I accumulation phenotype. What is the result of over-expressing Fam134b or Fam134c in Fam134a knockout MEFs?

GST-pull down experiments (Figure 1B) show that FAM1234A, FAM1234B, FAM1234C bind to LC3B while co-IP assays (Fig.EV6B, C) show the absence of interaction between FAM1234A and LC3B in cells. Authors should discuss this inconsistency. Also IgG controls in co-IP assays are missing (Fig.EV6C-E).

Are all FAM134 family members ubiquitously expressed in all tissues? Authors should clarify this in their discussion.

Minor points

1. N-terminal and C-terminal labels should be included in Figure 1A.
2. CNX label is missing in top picture of Figure 1C.
3. Invalid citation in page 7, check format.
4. Table 1 is missing.

5. Expanded view figure 6 A legend: did authors mean figure 7B when they referred to pro-collagen I western blot bands?
6. Expanded view figure 6 D: did authors mean figure 7G when they referred to pro-collagen I western blot bands?
7. Expanded view figure 6 E: did authors mean figure 7I when they referred to pro-collagen I western blot bands?

Rebuttal Letter

We thank the Referees for their insightful comments and suggestions. We performed several new experiments to address the open questions and included the new data in reshaped figures and text accordingly.

Referee #1:

In this manuscript, Reggio and co-workers have identified two new ERphagy receptors, FAM134A and FAM134C, both being similar to the already described FAM134B in that they contain a LIR motif that is required for their ERphagy function. However, in contrast to FAM134B, which induces ERphagy upon overexpression, FAM134A and -C mediated ERphagy require an activation signal. All FAM134s were found to interact with Calnexin and affect pro-collagen I levels, and although they have somewhat redundant functions when it comes to regulation of collagen I turnover, FAM134A seems to regulate this in an LIR-independent manner.

The results presented in this manuscript are generally well done, but unfortunately not of sufficient quality to support the conclusions drawn. The identification of FAM134A and FAM134C as ERphagy receptors is novel, but more insight into the mechanisms involved in regulation of their activity and how this differ from FAM134B under various (patho)physiological conditions are required to warrant publication in EMBO J. The authors should consider the specific comments below.

We thank the reviewer for her/his interest in our work and for her/his suggestions that we followed and included in the revised version as listed below in detail.

General comments:

1. The main shortcoming with this manuscript is the fact that all ERphagy data is based on overexpression of the different human FAM134s in U2OS cells (no knock-out or rescue), while their effects on collagen I levels are done in KO MEFs cells with rescue of mouse Fam134s. To be able to conclude they should do the ERphagy assays in KO-rescue cells (U2OS and/or MEFs).

While we can see the direction, the reviewer is pointing to, we believe that simply repeating all experiments in U2OS and MEFs cell lines is not adding further mechanistical insights to the data set. However, we performed additional experiments to complement our previous data obtained in U2OS and MEFs cells:

- We added mass spectrometry (MS) and microscopy analyses in U2OS to further conclude that FAM134s overexpression is responsible for ER degradation (Fig.3; Fig.EV3).
- We performed ER-phagy flux assays in FAM134 knockout and reconstituted MEFs (Fig.4).
- We performed additional cross-reconstitution experiments on *Fam134* knockout MEFs (Fig. 7).

Of note, FAM134s are quite ubiquitous expressed; however, different cell types have different ER structure and biological functions. MEFs produce and secrete Collagen I in larger amount compared to U2OS cells and have a higher basal autophagy flux. It is therefore unlikely to find identical substrates / effects in both cell lines.

2. When assessing effects of the various FAM134s on ERphagy, they generally use the overexpressed proteins themselves as a read-out (e.g. in Figure 2, EV3). It would be important to also quantify ERphagy with other ER markers to make sure they are not simply looking at different

rates of turnover of the FAM134 proteins. It is interesting that they did not have any western blot data of ER proteins with knockdown/overexpression of the FAMs, e.g. in EV5C, although it is a Co-IP experiment, it looks like Calnexin is not altered in the cells overexpressing the FAMs. Maybe they can also highlight ER resident proteins in their FAM KO mass specs to support Figure 2.

As the reviewer stated correctly in her/his next point (3), we used the KDEL reporter (general ER turnover) to analyze degradation of ER content. Therefore, we actually did use a readout independent of the various FAM134s, strongly suggesting we are not only looking at different rates of FAM134 protein turnover.

In the revised manuscript, we in addition employ a broader and unbiased approach to have a global overview of the effects of FAM134 overexpression in ER-phagy. We performed MS experiments on U2OS overexpressing cells where we analyzed the degradation rate of ER proteins after FAM134s overexpression and after ER-phagy induction. The new data clearly shows that FAM134s are functional in ER protein degradation in basal state and they further amplify ER degradation upon EBSS treatment (Fig. 3A-F). The ER-phagy flux experiments in ssRFP-GFP-KDEL U2OS overexpressing cells are in line with the MS dataset. Moreover, we validate our data showing, via IF, the CANX delivery to lysosomes upon FAM134s overexpression and ER-phagy induction (Fig.EV3). Our WB for SEC22B and REEP5 further support our hypothesis that FAM134s mediates degradation of (some) ER proteins (Fig.3I).

We also presented our MS data, from MEFs, in a different way. We now highlighted protein clusters that are significantly accumulated in knockout cells. One of the clusters includes ER resident proteins that are accumulated in all three *Fam134s* knockout cells. In addition, we performed the GFP-RFP-KDEL assays in knockout and reconstituted MEFs. All these data indicate that *Fam134* absence negatively affects ER turnover (Fig.4; Fig.5; Fig.EV4).

3. It is nice that they in Figure 3A use the ssRFP-GFP-KDEL ER-phagy reporter, but again only for cells overexpressing the different FAMs. They should do similar experiments with depletion of the different FAM134s +/- induction of ERphagy to show that these proteins are really required for the Torin-induced ERphagy. Assessing effects of overexpression, as done in Figure 3, is not very physiological relevant (could be overexpression artefacts). Rather they should do rescue experiments in a knock-out background.

As U2OS cells have a very low basal autophagy, we do not consider this cell line as optimal for a knockout/rescue experiment. Instead, we performed ER-phagy flux experiments in knockout and reconstituted MEFs. We see exactly the opposite effect than upon overexpression, supporting our understanding that we do not look at overexpression artefacts. As suggested by the referee we employed Torin1 to induce ER-phagy and we also performed the same experiment with the ER-stressor Thapsigargin (Fig.4).

Moreover, our MS experiments in knockout MEFs show that, in the absence of one of the *Fam134* members, there is an accumulation of ER proteins respect to the wild type cells (Fig.5; Fig.EV4). On the contrary, in U2OS cells overexpressing FAM134, we observed a decrease in the intensity of the ER proteins (Fig.3).

4. Based on in silico work with the RHD domains of the FAM134s, the authors propose a model where the RHDs of FAM134B and FAM134C are more dynamic than the RHD of FAM134A, resulting in differences in dynamics and curvature induction capacities of the respective RHDs, thus providing an explanation for why FAM134B and to a lesser degree C) are better at causing ER fragmentation. This is nice, but as there is no experimental evidence directly supporting this model (e.g. mutation in the respective RHDs), this is highly speculative.

We agree with the reviewer that our *in silico* data is mathematical modeling without experimental evidence and therefore speculative. We rewrote parts of the manuscript to explain this better. At the same time, we are confident that this model provides valuable insights into a possible regulation/activation. This specific model has been employed before and experimentally validated for FAM134B (Bhaskara et al., Nat Commun. 2019; Curvature induction and membrane remodeling by FAM134B reticulon homology domain assist selective ER-phagy; Siggel et al., J Phys Chem Lett 2021; FAM134B-RHD Protein Clustering Drives Spontaneous Budding of Asymmetric Membranes)

5. To be able to conclude that the various FAM134s have a different role in basal vs starvation-induced ERphagy it is important that the expression levels of the various FAM134 proteins are similar. How do the levels of HA-FAM134s when expressed in U2OS cells compare to each other and to endogenous FAM134 levels? In fig. EV5, one can see some differences in the expression levels of the different HA-FAM134s wt and LIR mutants. More worrying is the presence of several extra bands for the HA-FAM134s that can be seen in EV5 (especially upon high exposure E), where some do not correspond to the bands seen in EV5C,D. Why is the MW of FAM134B different in these Ips??

We agree with the reviewer that different levels of overexpression may influence the outcome of an experiment. In generating the different U2OS cell lines we started from the same parental cells and we cloned FAM134s CDNA in the same viral backbone. The IF images in Fig EV2B (basis for our statistics) have been taken with equal laser power. As the reviewer may acknowledge, the expression levels of FAM134A,B,C wild-type and FAM134A,B,C LIR mutant are very similar to each other. Therefore, we are confident that our data are correct. We additionally provided other WB to show that FAM134 expression level is comparable (not equal) in the different cell lines (Fig.EV5B). Differences in expression levels may as well be a consequence of FAM134 wild type being an active autophagy receptors and therefore constantly degraded.

Several bands of HA-FAM134 proteins: all autophagy receptors undergo degradation together with their cargo. We expect the extra bands to be intermediates of this degradation process, enriched during the IP of HA-FAM134. In our hands a rather common phenomenon. This is supported by the fact that the input shows a similar pattern as the IP (Fig.EV5B-D).

6. Page 5: the heading "FAM134A and FAM134C can drive ER fragmentation" is somewhat misleading, as they do not seem to drive fragmentation on their own. I would rather write ""FAM134A and FAM134C are required for starvation-induced ER fragmentation". Torin and starvation may not be the most specific ER-phagy inducers, maybe they can also try cyclopiazonic acid (CPA).

We edited the headline.

We performed ER-phagy flux experiments in presence of different concentrations of Thapsigargin (like CPA an inhibitor of SERCA). The results were comparable to results obtained with Torin1 (Fig.4C-E).

7. Since overexpression of FAM134A and C did not induce ER-phagy without starvation, a signal (like PTM) is probably required to activate these receptors. Are these proteins modified by phosphorylation or ubiquitination?

We agree with the reviewer that PTMs are most likely the cause of the switch between non-active and active forms of FAM134A and FAM134C. However, the simple fact that proteins are modified by PTMs is just a first step. In order to have an interesting and meaningful dataset, comprehensive validation of such PTMs would be needed. We believe that these analyses exceed the scope of this manuscript. Therefore, we decided to keep this point purely on the discussion level.

8. The authors argue that Fam134c seems to act as a facilitator of Fam134b activity in the degradation of misfolded collagen, while Fam134a mediates a parallel LIR-independent degradation pathway. This is based on largely descriptive data and more data is needed to conclude about the mechanisms involved, especially as the role of FAM134A in shaping ER morphology requires a functional LIR motif.

The data are also a bit confusing; in EV6A, it appears that the FAMs are not redundant as depletion of any one of the FAMs led to increase in collagen accumulation. However, in EV6D, Fam134a was able to rescue from collagen accumulation in Fam134b knockout, suggesting that Fam134a and Fam134b are redundant (in contrast to EV6A).

Single knockout of individual Fam134 genes leads to a strong ER phenotype in each of the three genotypes and proteomes of individual *Fam134* knockout MEFs differ. Therefore, by definition, individual Fam134 genes are not redundant. At the same time, the proteins do seem to have some overlapping/shared functions. Our MS data identifies a common set of proteins that are up- / down-regulated in the three *Fam134* knockout MEFs / U2OS cells overexpressing individual FAM134 proteins. Among proteins identified in MEFs two major clusters were detected: collagens and ER proteins. In the new Fig.3 and 5, we re-modelled the presentation of our MS data to better clarify analogies and differences among the Fam134s.

The final outcome for the clearance of Collagen I is the same - accumulation, therefore this is a shared function of Fam134s. However, while endogenous levels of Fam134a protein are insufficient to cope with the loss of Fam134b (thereby indicating that Fam134a is not just a back-up system, but fully 'busy' performing its own specific tasks), adding additional 'task-free' Fam134a protein to the system, by overexpression, can rescue the Fam134b phenotype of Collagen I accumulation. On the contrary, neither Fam134c endogenous protein levels nor overexpressed Fam134c protein is sufficient to rescue the Fam134b knockout Collagen I phenotype.

Collagen I is a common substrate, however not the only substrate of Fam134a. As such, action of Fam134a in this pathway (LIR independency) cannot be generalized. Some of the other functions of Fam134a are clearly LIR dependent as indicated by the ER enlargement phenotype that cannot be rescued by the LIR mutant.

9. In general, the figures should be more or less self-explanatory. This is not always the case here and the authors should pay specific attention to this. E.g. in figure 1C, they should write HA-FAM134 above image so that it is clear that HA in inset refers to overexpressed (not endogenous) FAM134

We went through the manuscript and edited Figures and Figure legends to make this information easy to catch for readers.

10. Figure legends should describe what was done and not the results, as is sometimes done, e.g. in Figure 2B; "Wild-Type FAM134s specifically bind to GST-LC3B and GSTGABARAPL1 via their LIR domain. FAM134s do not interact with GST-Ubiquitin and GST-Tetra-Ubiquitin».

We edit Figure legends accordingly.

Specific comments:

1. Figure 1B demonstrates nicely that both FAM134A and -C contain a functional LIR motif, but why only test binding to LC3B and GABARAPL1? They should test binding to the other ATG8 proteins as well. The set-up and labelling of this figure are confusing. They should indicate in the figure that this is using myc-FAM134s and GST-LC3B/GAB-L1. The empty should be GST? What is the difference between the input samples to the right vs the left part of the figure? The Ponceau staining must be from one of the pulldowns? LC3B and GAB-L1 run at different sizes in the two Ponceau blots.

We performed new pull-down experiments where we employed all six mATG8s.

We also fixed the issues with the Figure labeling, input and molecular weights. All changes are reported in the new Fig.2A-C.

2. Figure 1C: labeling of CANX in upper red image missing. It would be better to show a small region of the merge image as insets in different colors (instead of zoom out of same cell(s)). Why use different cells and constructs for Figure 1B (transient transfection of myc-FAM134s in HEK293 cell) and C (stable inducible U2OS HA-FAM134s)?

We edited the Figure accordingly with the referee's requests.

3. Figure 2: A-B. label all small images with HA and LC3B. Indicate in merge image where inset is taken from. C. should write Basal+BafA1 and EBSS + BafA1 for quantifications. They need to indicate how many cells were used for quantification and from how many experiments.

We edited the Figure legend (new FigE,F) and better explained the quantification method we adopted.

4. Figure 3A: should indicate in figure what is red/green (ssRFP-GFPKDEL) and use better quality images. Scale bar is missing.

The main purpose of new Fig. 3G is to give an easy visual idea of the assay. The quality of images (obtained by 10x magnification) are amplified parts of the original data and therefore cannot be improved. We added two time-lapse movies (movie EV7,8) including time stamp and scale bars of the total view/analyzed data. These movies also give an impression of the number of cells analyzed per well/view (of note, data obtained by a total of three wells/views are averaged at each time point).

5. Figure 3B: the legend should describe how the quantifications were done, how many cells, statistics etc. Why is the GFP/RFP ratio at 0h not always 1.0 (varies from 0.8 to 1.2)? What is the girl on the a-axis indicating? It is not so clear how the data in C differ from B. Are there significant differences between FAM134A, -B and C-??

Already under basal conditions, part of the reporter may be within lysosomes (basal autophagy flux; GFP quenched). Because the level of basic autophagy flux slightly differs in all cell lines, the GFP/RFP (old figures) or RFP/GFP (new figures) ratio is also not the same. In addition, intensity (the actual readout) of the two fluorophores per molecule is not the same. We indicated in the M&M part how many cells are seeded per well. The newly added movies are suggestive of how

many cells (clearly more than 200 per time point) are behind the quantification. Since we are using a ratio-based quantification data obtained by different number of cells are comparable. We changed the basis to RFP/GFP as it provides the reader with a more intuitive presentation of the data: raise in number = higher autophagy flux.

Personal note of AS: The “little lady in black” is an expression and reminder (since my diploma thesis) that science, despite the seriousness of the topic and frustrations alongside puzzling research, is creative and a source of joy and happiness. The picture is not invasive and doesn't affect the quality of the data. You may find the little lady as well visiting other publications, upon others Stolz & Dikic (2018), Trends Cell Biol; Rogov et al. (2017), EMBO Rep, Fig. EV1; Stolz et al. (2017), EMBO J., Fig.1; Stolz & Dikic (2014), Mol Cell

6. In Figure 7A they present "quantification of cells with misfolded pro-collagen I accumulation in wild-type, Fam134a,b,c single knockout MEFs and FAM134a,b,c single knockout cells reconstituted with the wild-type or LIR mutant form of the respective Fam134.", but it doesn't say how this is done, by quantification of images as shown in EV5B? Quantification of collagen intensity or number of cells showing any collagen staining? How many cells, how many experiments?

We updated our figure legends and Methods to be more informative about the presented data.

7. There is no Figure 7J although described in legend.

Sorry, this was our mistake

8. The text could benefit from thorough editing for grammar.

We edited the text

9. The introduction is a bit too extensive.

We edited and shortened the introduction

Referee #2:

The authors compared the activity of FAM134A, B, C in ER-phagy using three different readouts, ER remodeling, collagen quality control and ER-membrane degradation. The major finding of this work is that FAM134A appears to function different from FAM134B and FAM134C, and surprisingly, independent of its interaction with LC3s.

This is potentially an interesting paper suitable in scope and importance for EMBO J. However, currently some of the data are not convincing enough to support their conclusions. Substantial revisions are necessary for publication in EMBO.

We thank the Referee for her/his interest in our work and for her/his positive advices.

Major points:

1. Because FAM134A, FAM134B and FAM134C may interact (through their RTN domains) or affect each other in ER-phagy regulation, the best experimental design to compare their activities would be performing the rescue assays by expressing each individual WT proteins or LIR-deletion mutants in the triple-knockout cells, or at least, in the triple-knockdown cells.

All three FAM134 proteins have their individual tasks (Please see new Fig.3 and new Fig.5) and we do not primarily intend to compare the activity of individual Fam134 proteins in this manuscript. Instead, we aim to characterize FAM134A and FAM134C as ER-phagy receptors.

We know that double deletion of Fam134 genes have additive effects in mice (unpublished data), therefore triple knockout would presumably add additional stress to the cell. We would then analyze mainly stress response effects rather than their individual function, which is our main aim.

2. It is intriguing that FAM134A mediates pro-collagen degradation without LC3 interaction. Does this mean that FAM134a promotes procollagen degradation independent of the canonical ER-phagy? what is the mechanism? In Figure 5D, both FAM134A Δ LIR and FAM134C Δ LIR failed to rescue ER-phagy activities. It would be contradictory to the major conclusion of this work, if FAM134a Δ LIR drives the degradation of procollagen via macroautophagy pathway. Macroautophagy-independent degradation of ER components has been proposed. The authors may test the possibilities from other perspectives.

Along the lines of the reviewer, we investigated the possible involvement of the secretory pathway and test if Collagen I secretion is affected. Our data show that the secretory pathway is not affected in *Fam134* knockout MEFs. Despite the accumulation of pro-Collagen I in the cytoplasm, secretion of mature Collagen I is functional in all three genotypes. Please see Fig.EV4H.

The final outcome for the clearance of Collagen I is the accumulation, therefore this is a redundant function of Fam134s. However, while endogenous levels of Fam134a protein are insufficient to cope with the loss of Fam134b (thereby indicating that Fam134a is not just a back-up system, but fully 'busy' performing its own specific tasks), adding additional 'task-free' Fam134a protein to the system, by overexpression, can rescue the Fam134b phenotype of Collagen I accumulation. On the contrary, Fam134c endogenous protein levels nor overexpressed protein are sufficient to rescue the Fam134b KO collagen I phenotype. Collagen I is a common substrate, however not the only substrate of Fam134a. As such, action of Fam134a in this pathway (LIR independency) cannot be generalized. Some of the other functions of Fam134a are clearly LIR dependent as indicated by the ER enlargement phenotype that cannot be rescued by the LIR mutant. This, per se, is not contradictory or exclusive. In case of clearance of Collagen-I, a scaffold function (e.g. connection to CANX) and an interaction to another LIR-containing protein might be essential. Identification of the precise mechanism, how Fam134a regulate Collagen I homeostasis (at the moment not even completely clear for the well-studied Fam134b protein), is of biological interest but may exceed the scope of this manuscript.

Minor points:

1. The statement that "Here, we introduce FAM134A/RETREG2 and FAM134C/RETREG3 as new ER-phagy receptors." in the abstract was inappropriate, because FAM134a and FAM134b were implied as the ER-phagy receptors in previous study(Khaminets et al., 2015).

We edited the sentence. However, the manuscript Khaminets et al., Nature 2015 only presented the interaction between FAM134A and FAM134C with the mATG8s. Binding mATG8s is not sufficient to characterize a protein as a (ER-phagy) receptor, but needs additional characterization of the protein.

2. The FAM134C marker standard is not consistent in Figure 5E.

We fixed this issue.

3. The resolution of immunofluorescence image should be improved. The co-localization of FAM134 and CANX in Fig1 was weak.

We provide new images (Fig. 1B, EV3).

4. In Fig2C, statistic results do not match the representative images. It is obviously that HA dots were fewer in FAM134C (WT) than FAM134A and FAM134B after treatment with EBSS.

We have now clarified in figure legends that quantification presented in new Fig.2E refers to images presented in new Fig.EV2B. Quantification has been performed in an automated manner using predefined settings for all three cell lines and treatments. Even though representing the same conditions, images of new Fig.2D may be misleading, as the readers eye is caught by LC3B decorated ER fragments (white dots). Number of those may be different and have not been quantified in our study. We also better explained the quantification methods in figure legend.

5. "Although the LIR dependent activity of FAM134A and FAM134C does affect ER membranes fragmentation, it did not significantly influence ER tubular structure (ER branching), neither under basal nor starvation conditions". After treatment with EBSS, FAM134A and FAM134C significantly fragmented ER, ER theoretically would be degraded by ER-phagy. Why ER branching stay unchanged? Fig2A and B should be stained with ER marker.

FAM134A & FAM134C are expressed all along the ER and not only in defined areas as FAM134B. Moreover, after treatment with EBSS a relatively small portion of the ER is degraded, therefore it is unlikely that the entire ER network will be affected in a way to determine its complete disassembly. In contrast, FAM134B is extremely active and with this also harms the cell by breaking down the ER-network.

6. Fig.EV3C, quality of EM image should be improved. Gold particle labeling for FAM134A was not specific.

ER: endoplasmic reticulum
Mito: mitochondria
AV: autophagic vesicles

We believe that the IEM shows specific labelling of FAM134A. The protein was found in proximity of ER membranes and inside the autophagic vesicles. Some amount of the protein is present also in the cytosol associated to the ER membranes. We provide here additional pictures to clarify the distribution of

FAM134A. Overall, IEM data are in agreement with the IF images.

7. Compounds that induce ER stress can be used to induce ER-phagy and detect whether FAM134s protein is degraded, including ER marker.

We performed and included ER-phagy flux experiments in presence of different concentrations of the ER stressor Thapsigargin (an inhibitor of SERCA). The results were comparable to results obtained with Torin1 (Fig. 3J, Fig. 4D,E).

8. In Fig5C, the effect of Fam134c *-/-* reconstituted with WT protein is better than Fam134a, but was contradictory to the statistic results.
And why not use Fam134b *-/-* as control?

We added Fam134b as control. The selected images are representative and in line with the statistical analysis.

9. "Moreover, as previously reported for Fam134b (Forrester et al., 2019), Fam134a and Fam134c co-immuno-precipitate with Canx (Fig.EV5CE). This suggest that Canx may be working as co-receptor between mis-folded procollagen and the Fam134 family of ER-phagy receptors". whether the degradation of pro-collagen is affected in Canx knockdown cell?

Pro-Collagen I degradation is indeed affected in *Canx* knockout MEFs. Please see the WB below. This data was also reported in Forrester et al., EMBO J 2019.

10. In Fig.EV6A, Fam134b KO and Fam134b KO reconstituted with deltaLIR, Fam134b KO reconstituted with WT and Fam134c KO reconstituted with WT, the statistical data of these two groups was contradictory to western blot bands in Fig7B.

We compared the quantification presented in previous Fig.EV6A (new:Fig7C) with the corresponding WB in previous Fig7B (new Fig.7B) and could not identify a contradiction between the two data sets. Of note, the arrangement of cell lines in the bar graph and the WB is different, which may have caused confusion.

11. In Fig7I, it was concluded that "Fam134a over-expression had a stronger effect compared to that of Fam134b". In fact, Fam134a overexpression was much higher than Fam134b in Fig7H.

We revisited our data and still are under impression that overexpression of Fam134b and Fam134a is comparable in its amount (please compare EV5I 2nd and 3rd lane and not to WT cells). We performed the experiment to best knowledge, using cells reconstituted with cDNAs cloned in the same viral vector.

Referee #3:

Reggio and co-authors delineate FAM134A/RETREG2 and FAM134C/RETREG3 as new ER-phagy receptors in this manuscript. The authors compare FAM134A and FAM134C with their family member FAM134B, a well-known ER-receptor, and show that FAM134A and FAM134C require an activation signal to induce ER fragmentation, while FAM134B is sufficient to induce massive ER fragmentation in baseline conditions. They show that the three FAM134 family members display discrete conformations and ability to induce vesicle formation. All Fam134 family members are required to maintain ER homeostasis and are involved in the regulation of pro-

collagen I levels. However, the authors reveal a specific function for FAM134a which exerts its function through LIR independent (misfolded collagen I control) and LIR dependent (ER morphology) mechanisms. Overall, the MS is clear and the conclusions that the authors draw are well supported by the data. Also, the characterization of FAM134A and FAM134C as new ER-phagy receptors is novel and brings new light into the field. However, there are some controls that should be included and some points that need to be reinforced.

We thank the Referee for her/his interest in our work and for the positive criticisms she/he brought to us.

Major points:

The authors claim that mutation of the LIR motif in FAM134A and FAM134C are sufficient to abolish FAM134A/C-LC3B interaction, thus supporting the presence of a functional, canonical LIR motif in FAM134A and FAM134C. However, there is a faint FAM134A band in the blot where LC3B- FAM134A Δ LIR possible interaction is analysed (Figure 1B right panel) This blot should be improved.

We provided new pull-down experiments in order to clarify this point (new Fig. 2B). Moreover, from aminoacidic analysis of FAM134A we could not identify another putative LIR domains.

Moreover, does empty refer to cells transfected with pGEX-4T-1 plasmid?

Yes, it is. We exchanged the label in the Figure

What is the band observed in Panceau image? Authors should annotate this better.

Bands in Ponceau imaged show GST-baits.

FAM134C images of EBSS condition showed in figure 2A do not match quantification showed in 2C.

Quantification shows a higher number of HA dots in FAM134C than in FAM134B overexpression while pictures do not reflect that. Maybe authors could point with an arrow what they mean by HA dots. Does the quantification graph show a representative experiment or a pool of different experiments? Has any statistical test been performed for this quantification?

We thank the referee for pointing this out. We have now clarified in figure legends that quantification presented in new Fig.2E refers to images presented in new Fig.EV2B. Quantification has been performed in an automated manner using predefined settings for all three cell lines and treatments. Even though representing the same conditions, images of new Fig.2D may be misleading, as the readers eye is caught by LC3B decorated ER fragments (white dots). Number of those may be different and have not been quantified in our study. We also better explained the quantification methods in figure legend.

Why Torin 1 treatment induces specifically ER-phagy? Please explain it briefly.

So far, there are no chemicals that specifically induce ER-phagy. Torin 1 induces autophagy in general. We used Torin1 to induce ER-phagy because in our hands it was also robustly induce ER-phagy flux. Moreover, Torin1 was already used by other groups to investigate ER-phagy (Liang et al., Cell 2020). Same results were obtained with the more specific ER stressor Thapsigargin (new Fig.3J, Fig4D,E).

In their proteomics analysis, can the authors detail whether they used all the proteins for hierarchical clustering and what metric and linkage methods they have used for the same? It is also possible to perform hierarchical clustering on the first 2 components of the PCA, to identify proteins defining each class for those components.

We thank the referee for this insightful suggestion. We elaborated our data following her/his advice and we now provided a more complete and clearer Fig 5. Details regarding the data analysis are reported in Results & Methods sessions and in Figure legends.

Single knockout MEFs, compensatory increase in protein levels of the remaining two FAM134s were not observed (Figure 5A). However, Fam134c^{-/-} MEFs seem to have lower levels of Fam1234b than WT MEFs. Could authors comment on that?

This was a loading artifact. We provided a more explicative WB in order to clarify this point (new Fig. 4A).

Then authors claim that at least in MEFs, all three Fam134 family members have equally important roles in maintaining proper pro-collagen I quality control. They should downregulate Fam123 members in another cell line, such as U2OS, and measure pro-collagen I levels by western blot and/or perform Col1a1+Lamp1 immunofluorescence to validate the mechanism in a second cell line and rule out the possibility of cell-specific effects.

We actually expect cell-specific effects and that individual FAM134s have different tasks in different cell types. U2OS cells are not really devoted to collagens production so it may be difficult to appreciate differences in this protein level. FAM134s are quite ubiquitous expressed; however, different cell types have different ER structure and biological functions. MEFs produce and secrete Collagen I in larger amount compared to U2OS cells and have a higher basal autophagy flux.

We performed additional experiments to complement our data obtained in U2OS and MEF lines.

- We added MS and microscopy analyses in U2OS to further conclude that FAM134s overexpression is responsible for ER degradation (Fig.3; Fig.EV3).
- We performed ER-phagy flux assays in FAM134 knockout and reconstituted MEFs (Fig.4).
- We performed additional cross-reconstitution experiments (Fig. 7).

As such, we provide sufficient evidence in at least two cell lines (U2OS and MEFs) that FAM134 family members are ERphagy receptors. We are therefore confident to not be misled by a cell-specific effect.

It seems that knockout MEFs reconstituted with Fam134a rescue completely pro-collagen I accumulation, while reconstitutions with Fam134b and Fam134c show a partial rescue, there is still some pro-collagen I accumulation (Fig.EV5B and Figure 7A). Authors should comment on that.

We noticed that knockout cells (even after selection) needed a long time to regain homeostasis. Accumulation of collagen I in Fam134a knockout cells was less compared to the other two genotypes. Therefore, cells were able to regain wild-type homeostasis relatively fast. The experimental set is sufficient to make our claim, that reconstitution with Δ LIR mutants of Fam134b and Fam134c is insufficient to restore Collagen I levels.

Over-expression of Fam134a in Fam134b and Fam134c knockout MEFs is sufficient to fully rescue the pro-collagen I accumulation phenotype.

What is the result of over-expressing Fam134b or Fam134c in Fam134a knockout MEFs?

We overexpressed Fam134b and Fam134c in Fam134a knockout MEFs. We investigated Collagen I protein levels and we could appreciate the rescue of the phenotype (Fig.7J-M).

GST-pull down experiments (Figure 1B) show that FAM1234A, FAM1234B, FAM1234C bind to LC3B while co-IP assays (Fig.EV6B, C) show the absence of interaction between FAM1234A and LC3B in cells. Authors should discuss this inconsistency. Also IgG controls in co-IP assays are missing (Fig.EV6C-E).

In the pull-down experiments of Fig.1 (new 2A,B), we transfected FAM134 cDNAs in HEK cells and we used purified GST-mATG8s as baits. For the Co-IP we immunoprecipitated over-expressed FAM134s from U2OS stable cell lines and we performed WB against endogenous LC3B. The experimental settings are different and in the case of GST pull down we are using a large amount of LC3B compared to the endogenous level, thereby promoting an interaction. This technical aspect should explain the differences: the GST pull down is a biochemical assay aiming to map the LIR domain in FAM134A, while the Co-IP is closer to reality within a cellular context. Of note, we could detect an interaction with the GABARAPs in Co-IP settings. Please see new Fig.EV5G.

Are all FAM134 family members ubiquitously expressed in all tissues? Authors should clarify this in their discussion.

We analyzed several murine tissues Fig EV1C.

Minor points

1. N-terminal and C-terminal labels should be included in Figure 1A.

We edited the figure

2. CNX label is missing in top picture of Figure 1C.

We edited the figure (new Fig.1B).

3. Invalid citation in page 7, check format.

We edited the text

4. Table 1 is missing.

We uploaded all the Tables as a separate files

5. Expanded view figure 6 A legend: did authors mean figure 7B when they referred to pro-collagen I western blot bands?

We fixed this

6. Expanded view figure 6 D: did authors mean figure 7G when they referred to pro-collagen I western blot bands?

We fixed this

7. Expanded view figure 6 E: did authors mean figure 7I when they referred to pro-collagen I western blot bands?

We fixed this

Dear Alexandra,

Thank you for the submission of your revised manuscript to EMBO reports. We have now received the reports from former referee #1 and #2, copied below.

As you will see, both referees are very positive about the study and request only minor changes to clarify text and figures.

From the editorial side, there are also a few things that we need before we can proceed with the official acceptance of your study.

1) Please provide the main and the EV figures as individual production quality files as .eps, .tif, .jpg (one file per figure).

2) Please update the references to the alphabetical Harvard style EMBO reports has adopted. The abbreviation 'et al' should be used if more than 10 authors are present, author names should not be capitalized. The year needs to be in brackets and page numbers listed. See also <https://www.embopress.org/page/journal/14693178/authorguide#referencesformat>
You can download the respective EndNote file from our Guide to Authors https://endnote.com/style_download/embo-reports/

3) Movies:

- The Movie callouts need correcting to 'Movie EV#' throughout the text.
- Please provide legends for the movies as README.txt file and zip the movie together with its legend. The .zip file is then uploaded.

4) Tables: Please rename Table 1-5 to "Table EVx"

5) Datasets:

- Please add their name "Dataset EVx" to the first line in the legends tab.
- Dataset EV3: In the Analysis tab, the title says "Table 6". Please correct this.
- Dataset EV4: The Enrichment tab has "Table 7" in the title, please correct this.

6) Data availability section: Please provide links that resolve directly to the individual datasets deposited to PRIDE instead of the general link provided now.

Please see also this link describing the suggested format of this section:

<https://www.embopress.org/page/journal/14693178/authorguide#dataavailability>.

Data availability

- 7) Figure EV2A legend: Please define whether n=3 refers to biological or technical replicates.
- 8) Expanded View figure legends: Please correct "Expanded View X" to "Expanded View Figure X"
- 9) Please describe your findings in the Abstract in present tense.
- 10) Please complete the information on funding in our online submission system.
- 11) Our routine image analysis identified potential splice sites in the Ponceau blots shown in Fig. EV1C and in the LC3 Western blot shown in Fig. EV5F. Could you please double-check these panels. In case these blot images were spliced, please indicate this with a stippled line. Please also provide the source data for these blots.
- 12) Regarding the 'little lady in black' in Figure 3H: Thank you for clarifying the meaning of this little cartoon. I do agree with you that we should be reminded that science is creative and fulfilling despite all the frustrations that also come along with experiments. However, we are used to the fact that every item in a scientific figure has a meaning and is there for a scientific reason. In this respect the little cartoon is irritating and confusing as one is left searching for its meaning in the context of the data shown. I have discussed the matter once more, also with our chief editor, and we kindly ask you to remove it for the reasons above.
- 13) Finally, EMBO reports papers are accompanied online by A) a short (1-2 sentences) summary of the findings and their significance, B) 2-3 bullet points highlighting key results and C) a synopsis image that is 550x200-600 pixels large (width x height) in .png format. You can either show a model or key data in the synopsis image. Please note that the size is rather small and that text needs to be readable at the final size. Please send us this information along with the revised manuscript. Maybe this could be a place to reintroduce the lady in black?

With kind regards,

Martina

Referee #1:

The authors have generally done a good job addressing my initial comments and concerns, which has significantly increased the overall quality of the manuscript. There are however still a few minor points that have not been addressed in a satisfactory manner, as described below.

1. (old minor comment 3); in new Figure 2D they need to indicate the region of the inset. It is also strange to have a zoomed out picture of the same cell to the left...This is also the case in several

EV figures (e.g. EV2B,E)

2. (old minor comment 5); inclusion of the "Little lady» is fine, but with no description of what it means, it can rather be confusing for the reader (not everyone has read previous papers)....

3. (old minor comment 6); In Figure 7A they need to indicate in the legend what the data represent, how is the misfolding quantified? Quantification of collagen intensity or number of cells showing any collagen staining?

Referee #2:

The authors have addressed the concerns of the reviewer adequately.

Rebuttal to Referees' comments

Referee #1:

The authors have generally done a good job addressing my initial comments and concerns, which has significantly increased the overall quality of the manuscript. There are however still a few minor points that have not been addressed in a satisfactory manner, as described below.

1. (old minor comment 3); in new Figure 2D they need to indicate the region of the inset. It is also strange to have a zoomed out picture of the same cell to the left...This is also the case in several EV figures (e.g. EV2B,E)

We thank the referee for pointing this out. Now, we have indicated the regions of the insets for figure 2D and for all other images with zoomed views.

In some cases, zoomed out pictures of each single-channel of fluorescence micrographs have been included to have a comprehensive overview for those cases with a complex micrograph staining.

2. (old minor comment 5); inclusion of the "Little lady» is fine, but with no description of what it means, it can rather be confusing for the reader (not everyone has read previous papers)...

We apologize with this referee for causing confusion with the little lady. The "miss" on the axis is a mark of our lab, a sort of scientific lucky charm. We agree to this referee that, in the absence of opportune explanation, its present can cause confusion for readers. Now, we have removed the lady from the figure, accordingly.

3. (old minor comment 6); In Figure 7A they need to indicate in the legend what the data represent, how is the misfolding quantified? Quantification of collagen intensity or number of cells showing any collagen staining?

We thank the referee for pointing this out. As the referee noted, we have quantified the fraction of cells that are positive for collagen staining. Now we have rephrased the sentence of the caption accordingly.

Referee #2:

The authors have addressed the concerns of the reviewer adequately.

We thank the referee for the throughout review and insightful comments on our manuscript in the first round of revision. Thanks to his/her suggestions, we have now significantly increased the quality of our data and substantially strengthen the main messages of our work.

Rebuttal to editorial comments

From the editorial side, there are also a few things that we need before we can proceed with the official acceptance of your study.

1) Please provide the main and the EV figures as individual production quality files as .eps, .tif, .jpg (one file per figure).

We uploaded Main and EV Figures as .tif files

2) Please update the references to the alphabetical Harvard style EMBO reports has adopted. The abbreviation 'et al' should be used if more than 10 authors are present, author names should not be capitalized. The year needs to be in brackets and page numbers listed. See also <https://www.embopress.org/page/journal/14693178/authorguide#referencesformat>
You can download the respective EndNote file from our Guide to Authors https://endnote.com/style_download/embo-reports/

References are formatted using “Harvard” style EMBO Reports from EndNote

3) Movies:

- The Movie callouts need correcting to 'Movie EV#' throughout the text.
- Please provide legends for the movies as README.txt file and zip the movie together with its legend. The .zip file is then uploaded.

We made a .zip folder as requested

4) Tables: Please rename Table 1-5 to "Table EVx"

5) Datasets:

- Please add their name "Dataset EVx" to the first line in the legends tab.
- Dataset EV3: In the Analysis tab, the title says "Table 6". Please correct this.
- Dataset EV4: The Enrichment tab has "Table 7" in the title, please correct this.

We fixed

6) Data availability section: Please provide links that resolve directly to the individual datasets deposited to PRIDE instead of the general link provided now.
Please see also this link describing the suggested format of this section: <https://www.embopress.org/page/journal/14693178/authorguide#dataavailability>;

Data availability

- RNA-Seq data: Gene Expression Omnibus GSE46843

(<https://www.ncbi.nlm.nih.gov/geo/query/acc.cgi?acc=GSE46843>)

- [data type]: [name of the resource] [accession number/identifier/doi] ([URL or identifiers.org/DATABASE:ACCESSION])

We edited the Methods accordingly

7) Figure EV2A legend: Please define whether n=3 refers to biological or technical replicates.

We fixed

8) Expanded View figure legends: Please correct "Expanded View X" to "Expanded View Figure X"

We fixed

9) Please describe your findings in the Abstract in present tense.

We fixed

10) Please complete the information on funding in our online submission system.

We completed

11) Our routine image analysis identified potential splice sites in the Ponceau blots shown in Fig. EV1C and in the LC3 Western blot shown in Fig. EV5F. Could you please double-check these panels. In case these blot images were spliced, please indicate this with a stippled line. Please also provide the source data for these blots.

We fixed the issue in EV1C and we provided original source data for all WB

12) Regarding the 'little lady in black' in Figure 3H: Thank you for clarifying the meaning of this little cartoon. I do agree with you that we should be reminded that science is creative and fulfilling despite all the frustrations that also come along with experiments. However, we are used to the fact that every item in a scientific figure has a meaning and is there for a scientific reason. In this respect the little cartoon is irritating and confusing as one is left searching for its meaning in the context of the data shown. I have discussed the matter once more, also with our chief editor, and we kindly ask you to remove it for the reasons above.

We moved the black lady from Figure 3H

13) Finally, EMBO reports papers are accompanied online by A) a short (1-2 sentences) summary of the findings and their significance, B) 2-3 bullet points highlighting key results and C) a synopsis image that is 550x200-600 pixels large (width x height) in .png format. You can either show a model or key data in the synopsis image. Please note that the size is rather small and that text needs to be readable at the final size. Please send us this information along with the revised manuscript. Maybe this could be a place to reintroduce the lady in black?

We provided these information

Manuscript number: EMBOR-2020-52289V3

Title: Role of FAM134 paralogues in endoplasmic reticulum remodeling, ER-phagy and Collagen quality control

Author(s): Alessio Reggio, Viviana Buonomo, Rayene Berkane, Ramachandra Bhaskara, Mariana Tellechea, Ivana Peluso, Elena Polishchuck, Giorgia Lorenzo, Carmine Cirillo, Marianna Esposito, Adeela Hussain, Antje Hübner, Christian Hübner, Carmine Settembre, Gerhard Hummer, Paolo Grumati, and Alexandra Stolz

Dear Alexandra,

Thank you for supplying source data for all Western blots. Its analysis resulted in several points that still need your attention, before we can officially proceed with publication:

- 1) Table EV4: Please add the full references for the plasmids. Kirkin et al 2009 e.g. is not part of the main reference list and having more than author name et al is helpful in finding the appropriate reference
- 2) Please split the source data into one file per figure.
- 3) Source data for Fig. 3I, Vinculin appears not to match all blots in the Figure panel. Please double-check.
- 4) Source data for Fig. 4B, Vinculin, also appears different from the blots shown in the panel. The same applies to FAM134a and FAM134b LIR. Could you please doublecheck?
- 5) Source data for Fig. EV1C, FAM134C: the red box needs to be shifted down a bit.
- 7) Source data and Ponceau staining: we note that the Ponceau stainings were done on different blots and not on the membrane used for Western blot. In addition, we noted that one representative Ponceau staining is shown in most figures. I strongly recommend to indicate this in the respective figure legends. E.g. "A representative Ponceau staining is shown" for Fig. 2A-C and EV2A.
Or "An independent Ponceau staining is shown as loading control" for Fig. EV1C to make it clear that you did not stain the same membrane.
- 8) Thank you for supplying the source data for Fig. EV1C. The Ponceau-stained membrane seems to have been cut with a scalpel. If you wish to avoid any future ambiguities, you could indicate this splice with a stippled line.
- 9) Figure 3A: It is fine to insert the little lady in black in the schematic drawing. But we noticed that the cartoon depicting the experimental workflow is actually not very clear by looking at panel A. May I suggest modifying this panel? It would be better to move the legend on the right over to panel B, because for A the colors do not match the color of the cells, and the legend is anyway more relevant for panel (B). In addition, it is unclear from just looking at the graph whether the EBSS treatment was applied to -Dox AND +Dox cells or if the conditions were rather: -Dox under normal conditions, -Dox under EBSS conditions, +Dox under normal conditions? You could remove the pipet tips and expand the diagrammatic plan to illustrate all conditions. Could you please also put the lady in a good proportion to the other items in the cartoon? Currently it is disproportionately

larger than all other items. Thank you very much.

Once you have made these minor revisions, please use the following link to submit your corrected manuscript:

Link Not Available

If all remaining corrections have been attended to, you will then receive an official decision letter from the journal accepting your manuscript for publication in the next available issue of EMBO reports. This letter will also include details of the further steps you need to take for the prompt inclusion of your manuscript in our next available issue.

Thank you for your contribution to EMBO reports.

Kind regards,
Martina

Manuscript number: EMBOR-2020-52289V3

Title: Role of FAM134 paralogues in endoplasmic reticulum remodeling, ER-phagy and Collagen quality control
Author(s): Alessio Reggio, Viviana Buonomo, Rayene Berkane, Ramachandra Bhaskara, Mariana Tellechea, Ivana Peluso, Elena Polishchuck, Giorgia Lorenzo, Carmine Cirillo, Marianna Esposito, Adeela Hussain, Antje Hübner, Christian Hübner, Carmine Settembre, Gerhard Hummer, Paolo Grumati, and Alexandra Stolz

Dear Alexandra,

Thank you for supplying source data for all Western blots. Its analysis resulted in several points that still need your attention, before we can officially proceed with publication:

1) Table EV4: Please add the full references for the plasmids. Kirkin et al 2009 e.g. is not part of the main reference list and having more than author name et al is helpful in finding the appropriate reference

We added this reference in the reference list

2) Please split the source data into one file per figure.

We uploaded all the source data as separate files

3) Source data for Fig. 3I, Vinculin appears not to match all blots in the Figure panel. Please double-check.

One Vinculin has a higher exposure, we now also added the higher exposure original picture.

4) Source data for Fig. 4B, Vinculin, also appears different from the blots shown in the panel. The same applies to FAM134a and FAM134b LIR. Could you please doublecheck?

We doublechecked our raw data. We reported the correct western blot. We tried to use the same greyscale in the main figure so this could explain why it may appear slightly different. It is just a matter of bright/contrast. You can see that the small imperfections of the western blots are the same for each panel we reported.

5) Source data for Fig. EV1C, FAM134C: the red box needs to be shifted down a bit.

We fixed

7) Source data and Ponceau staining: we note that the Ponceau stainings were done on different blots and not on the membrane used for Western blot. In addition, we noted that one representative Ponceau staining is shown in most figures. I strongly recommend to indicate this in the respective figure legends. E.g. "A representative Ponceau staining is shown" for Fig. 2A-C and EV2A.

Or "An independent Ponceau staining is shown as loading control" for Fig. EV1C to make it clear that you did not stain the same membrane.

We edited the figure legends

8) Thank you for supplying the source data for Fig. EV1C. The Ponceau-stained membrane seems to have been cut with a scalpel. If you wish to avoid any future ambiguities, you could indicate this splice with a stippled line.

We indicated the splice as requested

9) Figure 3A: It is fine to insert the little lady in black in the schematic drawing. But we noticed that the cartoon depicting the experimental workflow is actually not very clear by looking at panel A. May I suggest modifying this panel? It would be better to move the legend on the right over to panel B, because for A the colors do not match the color of the cells, and the legend is anyway more relevant for panel (B). In addition, it is unclear from just looking at the graph whether the EBSS treatment was applied to -Dox AND +Dox cells or if the conditions were rather: -Dox under normal conditions, -Dox under EBSS conditions, +Dox under normal conditions? You could remove the pipet tips and expand the diagrammatic plan to illustrate all conditions. Could you please also put the lady in a good proportion to the other items in the cartoon? Currently it is disproportionately larger than all other items. Thank you very much.

We edited the figure

Once you have made these minor revisions, please use the following link to submit your corrected manuscript:

<https://embor.msubmit.net/cgi-bin/main.plex?el=A6lj1LD4A4COPx5J6A9ftd57GpHn1H2AiflnozshbAngY>

If all remaining corrections have been attended to, you will then receive an official decision letter from the journal accepting your manuscript for publication in the next available issue of EMBO reports. This letter will also include

details of the further steps you need to take for the prompt inclusion of your manuscript in our next available issue.

Thank you for your contribution to EMBO reports.

Kind regards,
Martina

Dr. Alexandra Stolz
Institute for Biochemistry II
Goethe University School of Medicine,
Theodor-Stern-Kai 7
Frankfurt am Main D-60590
Germany

Dear Alexandra,

Thank you for implementing the last minor changes. I am now very pleased to accept your manuscript for publication in the next available issue of EMBO reports. Thank you for your contribution to our journal.

At the end of this email I include important information about how to proceed. Please ensure that you take the time to read the information and complete and return the necessary forms to allow us to publish your manuscript as quickly as possible.

As part of the EMBO publication's Transparent Editorial Process, EMBO reports publishes online a Review Process File to accompany accepted manuscripts. As you are aware, this File will be published in conjunction with your paper and will include the referee reports, your point-by-point response and all pertinent correspondence relating to the manuscript.

If you do NOT want this File to be published, please inform the editorial office within 2 days, if you have not done so already, otherwise the File will be published by default [contact: emboreports@embo.org]. If you do opt out, the Review Process File link will point to the following statement: "No Review Process File is available with this article, as the authors have chosen not to make the review process public in this case."

Please note that under the DEAL agreement of German scientific institutions with our publisher Wiley, your paper might be eligible for open access publication in a way that is free of charge for the authors. Please contact either the administration at your institution or our publishers at Wiley (emboreports@wiley.com) for further questions.
(see also <https://authorservices.wiley.com/author-resources/Journal-Authors/open-access/affiliation-policies-payments/institutional-funder-payments.html>).

Should you be planning a Press Release on your article, please get in contact with emboreports@wiley.com as early as possible, in order to coordinate publication and release dates.

Thank you again for your contribution to EMBO reports and congratulations on a successful publication. Please consider us again in the future for your most exciting work.

Kind regards,

Martina

Martina Rembold, PhD

Senior Editor
EMBO reports

THINGS TO DO NOW:

You will receive proofs by e-mail approximately 2-3 weeks after all relevant files have been sent to our Production Office; you should return your corrections within 2 days of receiving the proofs.

Please inform us if there is likely to be any difficulty in reaching you at the above address at that time. Failure to meet our deadlines may result in a delay of publication, or publication without your corrections.

All further communications concerning your paper should quote reference number EMBOR-2020-52289V4 and be addressed to emboreports@wiley.com.

Should you be planning a Press Release on your article, please get in contact with emboreports@wiley.com as early as possible, in order to coordinate publication and release dates.

Corresponding Author Name: Alexandra Stolz

Journal Submitted to: EMBO Rep

Manuscript Number: EMBOR-2020-52289V2